



# Development and validation of a global 1/32° surface wave-tide-circulation coupled ocean model: FIO-COM32

Bin Xiao[1,2,3], Fangli Qiao[1,2,3*], Qi Shu[1,2,3], Xunqiang Yin[1,2,3], Guansuo Wang[1,2,3], Shihong Wang[1,2,3]

[1]First Institute of Oceanography, and Key Laboratory of Marine Science and Numerical Modeling, Ministry of Natural Resources, Qingdao 266061, China
[2]Laboratory for Regional Oceanography and Numerical Modeling, Pilot National Laboratory for Marine Science and Technology, Qingdao 266237, China
[3]Shandong Key Laboratory of Marine Science and Numerical Modeling, Qingdao 266061, China

*Correspondence to*: Fangli Qiao (qiaofl@fio.org.cn)

**Abstract.** Model resolution and the included physical processes are two of the most important factors that determine the realism of ocean model simulations. In this study, a new global surface wave-tide-circulation coupled ocean model FIO-COM32 with a resolution of 1/32°×1/32° is developed and validated. Promotion of the horizontal resolution from 1/10° to 1/32° leads to significant improvements in the simulations of surface eddy kinetic energy (EKE), main paths of the Kuroshio and Gulf Stream, and the global tides. We propose the Integrated Circulation Route Error (ICRE) as a quantitative criteria to evaluate the simulated main paths of Kuroshio and Gulf Stream. The non-breaking surface wave-induced mixing (Bv) is proven to still be an important contributor that improves the agreement of the simulated summer mixed layer depth (MLD) and the Argo observations even with a high horizontal resolution of 1/32°. The mean error of the simulated mid-latitude summer MLD is reduced from -4.8 m in the numerical experiment without Bv to -0.6 m in experiment with Bv. By including the global tide, the global distributions of internal tide can be explicitly simulated in this new model and are comparable to the satellite observations. Based on Jason3 along-track sea surface height (SSH), wave number spectral slopes of mesoscale ranges and wave number-frequency analysis show that the unbalanced motions induced SSH undulation is a key factor for the substantially improved agreement between model and satellite observations in the low latitudes and low EKE regions. For ocean model community, surface waves, tidal currents and ocean circulations have been separated into different streams for more than half a century. This paper suggests that it should be the time to merge these three streams for new generation ocean model development.

# 1 Introduction

As the computing power has been increasing rapidly, approximately one order of magnitude higher with every five years, the state-of-the-art computing ability for modern global ocean numerical models is becoming enormously high, leading to recent achievements in high resolution global ocean models. The definition of "high resolution" of current stage





global ocean models may refer to those with horizontal resolutions ranging from 1 to 5 km, which are well beyond the mesoscale resolving threshold in most open oceans (Hallberg, 2013). Further improved resolution has a significant impacts on the simulated eddy activities (Thoppil et al., 2011; Sasaki and Klein, 2012; Biri et al., 2016; Ajayi et al., 2020), the

vertical mass and buoyancy fluxes (Capet et al., 2016; Su et al., 2018; Dong et al., 2020), and even the representation of large scale circulations (Lévy et al., 2010; Chassignet and Xu, 2017). These high resolution models are not only helpful in obtaining new scientific understanding on mesoscale, sub-mesoscale, eddies and mixed layer dynamics, but are also very useful in evaluating the satellite products of both current (Amores et al., 2018) and future generations, such as SWOT projects. Table 1 summarises recent developments in high resolution global ocean models. Not only is the model horizontal

resolution significantly increased, which now ranges from 1/20° to 1/48°, but the ocean model physical processes in ocean models are also making notable progresses. For example, HYCOM25 and LLC4320 are global tide-circulation coupled models, and FIO-COM32 is a global surface wave-tide-circulation coupled model.

**Table 1 Recent developments in global high resolution ocean models**

| High resolution global models | Ocean model names | Horizontal resolution | References |
|---|---|---|---|
| HYCOM25 | HYCOM | 1/25° | Savage et al., 2017a and b; Arbic et al., 2018 |
| LLC4320 | MITgcm | 1/48° | Rocha et al., 2016 |
| ORCA36 | NEMO4 | 1/36° | https://github.com/immerse-project/ORCA36-demonstrator |
| LICOM3-HIP | LICOM3 | 1/20° | Wang et al., 2021 |
| FIO-COM32 | FIO-COM-HR | 1/32° | This paper |

Improving the representation of physical processes in ocean models has been the most fundamental aspect for ocean general circulation model (OGCM) development. Since the establishment of the first OGCM (Bryan and Cox, 1967), surface wave models, tide models, ocean internal wave models and OGCMs have been separated into different streams (Mellor and Blumberg, 2004; Qiao et al, 2004). The most uncertainty term in all OGCMs is ocean turbulence. As a result, the vertical structures of ocean temperature and salinity cannot be accurately simulated or predicted. For example, the simulated mixed

layer depth (MLD) in the upper ocean is too shallow, and the sea surface temperature (SST) is overheating during the summer in nearly all OGCMs. Qiao et al (2004) proposed a upper ocean mixing scheme of $B_v$ which plays a dominant role in turbulent mixing of the upper ocean, analytically expressed $B_v$ as a function of the wave number spectrum which can be calculated from a wave model (Qiao et al., 2004, 2016). Ocean surface wave-circulation coupling has become one of the most important directions of ocean model development. $B_v$ has been widely adopted in a series of different OGCMs and

climate models, and all showed dramatic improvements (Qiao et al., 2004; Shu et al., 2010; Song et al., 2012; Fan and





Griffies, 2014; Wang et al., 2019). Bv is particularly effective in remedying the shallow biases of the simulated summer MLD and overestimated SST in summer. The following two additional points show surprising results: Firstly, after closing the traditional vertical turbulence schemes, only Bv can properly simulates the global ocean, which indicates that Bv plays dominant role in vertical mixing in the upper ocean (Qiao and Huang, 2012). Second, Bv can cause the MLD to become

shallower in winter in climate models, which is also an improvement for climate models (Chen et al., 2019). Although the effect of Bv has been comprehensively verified in many course resolution OGCMs, however, most of these numerical experiments were based on climatological simulations and validated against climatological observations. In order to investigate the impacts of horizontal resolution on the model MLD biases, taking the significantly different behaviours of course and high resolution models into consideration, it is still necessary to validate the effect of Bv in the high resolution

OGCM. For the first time, this paper investigate the Bv effects in high resolution OGCM of 1/32° and validated against in-situ observations.

Ocean tides have been recognised as a fundamental aspect regulating the hydrodynamic environments in shallow regions (e.g. Simpson and Hunter, 1974; Garrett and Loder, 1981; Holt and Umlauf, 2008; Lin et al., 2020); thus, ocean tides are often included in regional ocean models. The tidal mixing controls the formation of thermal fronts in coastal regions, and

generates upwelling (Lü et al., 2006, 2008, 2010). The regional surface wave-tide-circulation coupled model of the China Seas has shown excellent performance (Xia et al., 2006), and has been applied in operational ocean forecasting systems (Wang et al., 2016). In recent years, the high resolution global OGCM forced by a realistic atmospheric data begins to include ocean tide explicitly (Arbic, et al. 2010, 2012, 2018; Rocha et al., 2016). One of the benefits of including ocean tide in eddy-resolving resolution global OGCMs is that the global distribution of internal tide fields together with mesoscale

eddies can be resolved concurrently (e.g. Arbic et al., 2010, 2012, 2018; Buijsman et al., 2015; Shriver et al. 2012; Ansong et al., 2018; Timko et al., 2019). In addition, inclusion of ocean tide in a global 1/48° ocean model (MITgcm llc4320) leads to more realistic representation of the unbalanced and ageostrophic motions (Rocha et al., 2016).

To examine whether the high resolution OGCM can faithfully reproduce the ocean environment, the comparison of wave number/frequency spectra with observations is widely adopted (e.g. Sasaki and Klein, 2012; Richman et al., 2012;

Chassignet and Xu, 2017, hereafter CX17; Savage et al., 2017ab; Biri et al., 2018). Therefore, the wave number spectral slope in the 70-250 km mesoscale range becomes an important criteria for OGCMs' validation. Sasaki and Klein (2012, hereafter SK12), note that as the horizontal resolution improved to 1/30°, the wave number spectral slope in the 70-250 km mesoscale range of the model shows quite inconsistent patterns with that of satellite observations. The satellite observations show strong latitudinal variability (Xu and Fu, 2012), in high latitude and high eddy kinetic energy (EKE) regions, the slopes

range between -5 and -4 which is generally consistent with the theory of mesoscale turbulence. However, the slopes are much flatter in tropical and low EKE regions which is substantially distinct from the theory of mesoscale turbulence. In the high resolution models such as SK12 and CX17, the slopes range between -5 and -4 in both low and high latitude regions. More importantly, Chassignet and Xu (2021, hereafter CX21), showed that as the tide is included in their 1/50° Atlantic regional ocean model, the resolved internal tide in sea surface height (SSH) signals is the main reason in explaining the



inconsistency between model and satellite observations. Although ocean tides are included in regional OGCMs, there are still few attempts to include tides in global OGCMs.

This paper aims to answer the following three key questions through establishing a global 1/32° surface wave-tide-circulation coupled ocean model FIO-COM32: What are the main effects of increasing horizontal resolution from 1/10° to 1/32° in a global ocean model? What are the main effects of Bv in high resolution OGCM? Inspired by the results of an

Atlantic regional high resolution ocean model of CX21, what is the tidal effect in a global 1/32° high-resolution OGCM on simulating the wave number spectral slopes?

This paper is organised as follows. Section 2 describes the model configurations and designment of the numerical experiments. In Section 3.1, we present the results of basic aspects of the new global 1/32° surface wave-tide-circulation coupled ocean model, and illustrate the effects of increased horizontal resolution by comparing with a previous global 1/10°

model. Section 3.2 shows the effects of ocean surface wave and tide coupling in the global 1/32° ocean model. Section 4 provides a summary and discussion.

## 2. Model description and numerical experimental designment

### 2.1 Model description

The FIO-COM consists of a global ocean circulation model, a sea ice model, and a global ocean wave model. The

ocean component is based on the Modular Ocean Model 5 (Griffies, 2012), the sea ice component is based on the Sea Ice Simulator (Winton, 2000), and the surface wave component is based on the MASNUM (laboratory of Marine Sciences and Numerical Modelling of MNR of China) ocean wave model (Yang et al., 2005; Qiao et al., 2016). Currently, the variable exchanged between the ocean wave and the ocean circulation model is Bv. Bv is calculated in the ocean wave model following Qiao et al. (2004), as in Equation (1), $S(k)$ represents the surface wave number spectrum, and $k$ is the surface

wave number. As Bv is incorporated into the vertical mixing schemes of the ocean circulation model, $\alpha$ is a tuneable parameter for practical purposes, and we set it as 0.3 in this paper (Wang et al, 2010).

$$B_v = \alpha \int_k S(k) e^{-2|k|z} dk \cdot \frac{\partial}{\partial z} \left[ \int_k \omega^2 S(k) e^{-2|k|z} dk \right]^{\frac{1}{2}} \qquad (1)$$

The variable exchanged between the ocean wave and the sea ice models is sea ice concentration (SIC, from sea ice model to ocean wave model). The SIC is used to calculate time varying masks of the ocean wave model, the model domain

that has SIC exceeding 15% is masked out as land during the ocean wave model integration.

According to different research purposes, there are two different coupling strategies for the wave model component. Firstly, in the lower resolution numerical experiments, where the computational costs are not expensive, the wave model component is coupled online with the ocean circulation and sea ice models. The real-time online data exchanges are achieved based on the subroutine version of the MASNUM wave model. In this way, the wave model component could be coupled

with ocean circulation and ice components through direct calling of the MASNUM wave model as subroutines in the ocean





circulation model. The ocean wave, ocean circulation, and sea ice components share the same model grids. Secondly, in the numerical experiments with high computational costs, and with research focusing on ocean dynamics, the wave component can be turned off to save computational resources (the computational cost of the ocean-ice coupled model is approximately half that of the surface wave-ocean-ice model). In this configuration, the surface wave induced mixing coefficients are saved

in data files previously are then read into the OGCM.

The horizontal grid of FIO-COM32 is a tripolar grid (Murray, 1996) with a horizontal resolution of 1/32°×1/32°. The model covers the entire global ocean, with a latitudinal coverage of 82°S to 90°N and a longitudinal coverage of 280°W to 80°E respectively. Vertically, the z* coordinate (Adcroft and Campin, 2004) is adopted and the thickness is 2 m at surface and increases to 367 m at the bottom gradually. The vertical grid has 54 or 57 levels depending on whether ocean tide is

introduced. In the numerical experiments without ocean tide, the maximum depth is 5500 m. While the ocean tide is explicitly included, the maximum depth extends to 7000 m by including additional 3 bottom levels in OGCM. The maximum total grid size is 11520×5504×57. Model topography is from GEBCO (IHO-IOC, 2018) and is smoothed by applying a radial filter (Arbic et al., 2004). Model bottom cells are set as partial cells (Adcroft et al., 1997). The horizontal mixing scheme is a bi-harmonic operator with diffusive velocities of 1.96 cm/s for momentum and 0.65 cm/s for tracers respectively. Both the

1/10° and 1/32° models use the same model settings. The viscosity is calculated from diffusive velocity times the cube of the grid spacing, which means that the viscosity is more than 30 times smaller in the 1/32° model than that of 1/10° model (or simply called FIO-COM10). The vertical mixing scheme is the KPP scheme (Large et al., 1994) and Bv. The background vertical viscosity and diffusivity are set as $1.0×10^{-4}$ $m^2s^{-1}$ and $3.0×10^{-5}$ $m^2s^{-1}$ respectively. Bv is calculated from a global 1/4° online coupled FIO-COM and incorporated into both vertical diffusivity and viscosity. Neither sea surface temperature nor

salinity is restored to observational data. The bottom drag is quadratic with a coefficient of $2.5×10^{-3}$.

Ocean tide is explicitly included through introducing eight main tidal generating potentials including $M_2$, $S_2$, $N_2$, $K_2$, $K_1$, $O_1$, $P_1$, and $Q_1$ in the momentum equations following Schiller and Fiedler (2007). The treatment of self-attraction and loading (SAL) is a simple scalar approximation following Arbic et al. (2010), and the scalar alpha is set to 0.93. The SAL is applied only on the tidal elevations, thus a 25-hour running average time filter is adopted to get the slow-varying sea surface height

related with large scale ocean circulation. A topographic drag scheme (Jayne and St. Laurent, 2001) is introduced in the barotropic momentum equation, and this scheme is enabled in the barotropic numerical experiments but closed in the baroclinic experiments for that a large portion of the barotropic to baroclinic tidal energy conversion can be explicitly resolved in the baroclinic experiments.

The initial conditions including sea temperature, salinity, velocity and sea surface height (SSH) are interpolated from

outcomes of a global operational ocean forecasting system based on FIO-COM10 with horizontal resolution of 1/10° (Qiao et al., 2019; Shi et al., 2018; Sun et al., 2020). As a result, the new 1/32° model enters a steady state after a short spin-up period of about only one year (Figure A-1). The atmospheric forcing is from the Global Forecast System (GFS) of National Centers for Environmental Prediction (NCEP) (https://www.nco.ncep.noaa.gov/pmb/products/gfs/) with a horizontal resolution of 1/4°. The ocean surface heat and momentum fluxes are calculated via the bulk formula (Large and Yeager,





2004), and wind stress is calculated using relative wind.

## 2.2 Numerical experimental designment

To investigate to what extent the surface wave-tide-circulation coupled ocean modelling framework contributes to the newly established FIO-COM32 model, and taking the expensive computational cost into consideration, two numerical experiments are designed. In EXP1, only the OGCM is active, neither ocean surface wave nor tide is included, and the
simulated period is from 1 June of 2016 to 31 December of 2019. In EXP2, wave-tide-circulation fully coupling is enabled which means both Bv and tidal currents are activated. Since the online coupled surface wave model will almost double the computational cost compared with the ocean-only counterpart, in this paper the 1/32° numerical experiments reads in the Bv data which were calculated from a global 1/4° online coupled FIO-COM for the exactly same time period. Here we mainly focus on the effect of Bv in the new 1/32° ocean model, and due to the large-scale characteristics of the surface wave
simulations and the extremely expensive computational costs, we avoid to run a global 1/32° online coupled surface wave-ocean circulation coupled model directly.

EXP1 starts from outcomes of the global FIO-COM10 forecast system on 1 June of 2016. EXP2 branches from EXP1 on 1 July of 2017. The data analysis focuses on the period from 1 January 2018 to 31 December 2019. The model output frequency of three-dimensional variables is daily, and the output frequency of SSH and steric SSH is hourly. Additional two
experiments with a horizontal resolution of 1/10° is also conducted to investigate the effects of model resolution. The settings of the two 1/10° model are kept identical with that of EXP1 and EXP2 respectively, hence they are named as EXP1Low and EXP2Low. As has mentioned in the model description, neither sea surface temperature nor salinity is restored to observational data in all of the numerical experiments, the model drift during the experiment period is examined to ensure that the model maintains a reasonable evolution (Figure A-2). Since the time span of the numerical experiments is not long
(3.5 years), the model drift is generally quite small.

Another two numerical experiments, OGCM+Bv and OGCM+tide, should be conducted to clearly identify the exact effects of surface wave and tide respectively. However, considering the high computational costs, we only perform EXP1 and EXP2.

In order to understand how the model resolution affects the simulated batrotropic tide, two global barotropic tide
models with horizontal resolution of 1/10° and 1/32° are compared. Ocean temperature and salinity in the global barotropic tide models are set as uniform values, and are kept as constants during model integrations. The parameterized topography drag scheme is needed in the barotropic tide models since the energy transfer mechanism from barotropic to baroclinic tide is missing. As have been noted, a topography drag scheme (Jayne and St. Laurent, 2002) is incorporated into the momentum equation following previous works (Jayne and St. Laurent 2002; Egbert et al., 2004; Arbic et al., 2004), the drag coefficient
is best tuned for both numerical experiments. The model topography in this paper is different from our previous work (Xiao et al., 2016) in which the treatment of ice shelf topography of Antarctica is more realistic by using the water column thickness of BEDMAP2 (Fretwell et al., 2013), and this would yield smaller RMSE.





## 3 Results

### 3.1 Effects of horizontal resolution

In this section, the surface EKE and SSH simulated by the 1/32° models are compared with those of 1/10° models to investigate the effects of different horizontal resolution.

One of the most noteworthy facts is that the EKE increases significantly as the model horizontal resolution increases, as CX17 pointed out that this increase is due to smaller effective horizontal viscosity and more active mesoscale and sub-

mesoscale motions of higher resolution models. The simulated surface EKE and satellite observations are compared here to investigate the effects of improved horizontal resolution in FIO-COM32. The surface EKE is calculated as follows.

$$\text{EKE} = \frac{1}{2}\langle u'^2 + v'^2 \rangle \quad (2)$$

where $u'$ and $v'$ are anomalous zonal and meridional components of surface geostrophic velocity, respectively, which are calculated from sea level anomalies referenced to the mean sea level of 2018 and 2019. The bracket indicates the time

averaging of the model years of 2018 and 2019. Satellite derived surface EKE are calculated using absolute dynamic topography data of Ssalto/Duacs altimeter products produced by the Copernicus Marine and Environment Monitoring service (CMEMS, http://marine.copernicus.eu). Generally, increasing the model resolution from 1/10° to 1/32° significantly improves the agreement of the simulated EKE and satellite observations (Figure 1). Compared with the 1/10° model, the 1/32° model shows much enhanced EKE almost everywhere in the global ocean, especially in the regions of subtropic, mid-

latitudes, and western boundary current systems, where surface EKE in the 1/10° model is too weak against satellite data. This indicates the horizontal resolution of 1/10° is insufficient to properly resolve mesoscale and sub-mesoscale motions, while the simulations of the 1/32° model are much improved. For the Kuroshio and Gulf Stream extensions, not only the EKE amplitude is enhanced but also their spatial distribution structures such as the separation point and the extension range agree with the satellite observations much better in the 1/32° model. The calculated global mean EKE time series (Figure 2a)

also show improved agreement for the 1/32° model with the satellite observations, while the 1/32° model has higher global mean EKE values. In order to understand how does the small scale motions resolved by the 1/32° model contribute to the enhanced EKE level, a spatial filter with radius of ~50 km is applied to separate the large scale (mesoscale or larger) and small scale (some sub-mesoscale) motions. As a result, the output of SSH of the 1/32° model is low-pass filtered, then the corresponding low-pass EKE (Figure 2b) is calculated and the residuals of total EKE are regarded as small scale

contributions (Figure 2c). Both the low-pass model EKE and CMEMS are on the same 1/4° grid which makes them more comparable. The time series of the low-pass EKE of the 1/32° model (the thick solid red line in Figure 2a) is significantly lower than that of original 1/32° model and also lower than that of CMEMS, which indicates the sub-mesoscale motions (such as small scale eddies and fronts etc.) have important contributions to the enhanced EKE level of the 1/32° model





(Figure 2c). On the other hand, the low-pass EKE of the 1/32° model is still much more energetic than that of 1/10° model,
this indicates the mesoscale eddy fields are more energetic in the 1/32° model than those in 1/10° model. The calculated root
mean square error (RMSE) of the two-year-averaged EKE between model simulations and satellite observations decreases
from 63.5 $cm^2s^{-2}$ of the 1/10° model to 31.7 $cm^2s^{-2}$ of the 1/32° model. The RMSE is calculated for regions that have
water depths exceeding 1000 m and are located between 65°S and 65°N.

Both the mean SSH and SSH standard deviation (STD) of the 1/10° and 1/32° models are compared in Figure 3. The
large scale circulation patterns are reasonably simulated in both the 1/10° and 1/32° models (Figures 3a, c, e). For the
detailed simulation structures, such as pathways and separation points of the west boundary currents, the 1/32° model shows
significant improvements over those of the 1/10° model. As the SSH STD comparisons show, the 1/32° model is more
energetic than the 1/10° model in the well-known mesoscale active regions (Figures 3b, d, f), which has already been
discussed in the EKE analysis. In the Kuroshio region, the 1/10° model shows over concentrated energy near the separating
point and the Kuroshio Extension is too weak to penetrate the interior of the Pacific Ocean, while these characteristics are
much improved in the 1/32° model.

To better understand the effect of model resolution on the large scale circulations, we investigate the simulated main
paths of the Kuroshio and Gulf Stream. The annual mean SSH of 1/10° (EXP2Low) and 1/32° (EXP2) models of 2019 are
compared with the corresponding altimeter observations of CMEMS. In this paper, we propose the Integrated Circulation
Route Error (ICRE) as a quantitative criteria to assess the quality of the simulated paths of Kuroshio and Gulf Stream. The
ICRE is calculated as follows: Firstly, the path of the western boundary current is defined as the contour edge of SSH at
level of 0.16 m. Note that the regional averaged value is extracted from the SSH before the ICRE analysis is conducted to
ensure the SSH of the model and observation are comparable. Secondly, the ICRE is calculated as the total integrated misfit
area between the contour edge of the model and CMEMS.

As the resolution increases from 1/10° to 1/32°, the simulated Kuroshio path is significantly improved (Figure 4), the
ICRE is decreased from $3.01x10^{11}$ m² to $1.73x10^{11}$ m². The most notable improvement is that the 1/32° model is able to
reproduce the Kuroshio large meanders, while the 1/10° model can not. The ICRE of the Gulf Stream is decreased from
$4.85x10^{11}$ m² of 1/10° model to $3.27x10^{11}$ m² of 1/32° model. We should note that for the comparisons of the Gulf Stream
region, the re-circulation part of the ICRE is masked out to focus on the simulation of main path of Gulf Stream. Compared
to the 1/32° model, the 1/10° model is not able to reproduce the deep penetration of the Gulf Stream into the Atlantic ocean
(Figure 5).

Figures 6 and 7 show the simulated relative vorticity of sea surface current of the 1/10° and 1/32° models. They clearly
show that, as the horizontal resolution increases, the model can resolve much more sub-mesoscale and mesoscale motions.
The 1/32° model also shows obviously more significant seasonal variations over the 1/10° model, this is because the mixed
layer instabilities are substantially enhanced in the 1/32° model, especially during winter, this phenomenon is consistent with
previous publications (e.g. Sasaki et al., 2017; CX17; Dong et al., 2020). In the higher resolution model, the small-scale eddy
activities is greatly enhanced during winter compared to that of summer, while this is not as obvious in 1/10° model. Since





the horizontal scales of the resolved fronts become much smaller in 1/32° model, the potential effects of explicit Stokes shear force are estimated and compared among the models with different horizontal resolutions. The estimation is based on the

parameter $\epsilon$ proposed by McWilliams and Fox-Kemper (2013):

$$\epsilon = \frac{V^s}{fH^s}\frac{H}{L}$$

$V^s$ is the Stokes drift characteristic velocity, $f$ is the Coriolis parameter, $H$ and $L$ are the depth and width of the front, and $H^s$ is the Stokes drift decay depth. The estimation is made for a typical section in Kuroshio extension region, which is marked as a black line in Figure 6. Figures A-3 and A-4 show the evolution of relative vorticiy and temperature along a typical section in Kuroshio extension region. The typical horizontal scale of resolved fronts in 1/10° model is about 60 km,

and the horizontal scale in 1/32° model is about 3 times smaller ~20 km. The vertical scales of the resolved fronts of both 1/10° and 1/32° model are quite similar, the summer (winter) has a vertical scale of about 30 m (300 m). The dimensionless parameter representing the wave parameters in $\epsilon$ is $\frac{V^s}{fH^s}$, which is typically O(10-100) (Suzuki et al., 2016). The major difference of the estimations between the 1/10° and 1/32° models is the frontal aspect ratio in $\epsilon$, which is $\frac{H}{L}$. During summer, the frontal aspect ratio is estimated to be ~1/1000 and ~3/1000 in 1/10° and 1/32° models respectively, which yield the

estimated $\epsilon$ to be ~1/10 and ~3/10. While during winter, the frontal aspect ratio is estimated to be ~1/200 and ~3/200 in 1/10° and 1/32° models respectively, yielding the estimated $\epsilon$ to be ~1/2 and ~3/2. Hence, it can be concluded that as the horizontal resolution increased from 1/10° to 1/32°, the critical resolution may be reached at least during winter time that the explicit Stokes shear force is not to be neglected. In this paper, the Bv is included and could be treated as a "bulk" mixing term accounting for these wave-turbulence interactions. The effect of explicit implementation of surface wave induced forces

in the 1/32° model remains to be explored in the future.

Reasonable global tide simulation is an important prerequisite that tide and OGCM could be coupled together, since tuning global barotropic tide can also yield better model topography settings, especially that some numerically unstable topography features can be fixed through this practice. The accuracy of simulated global barotropic tide is quite sensitive to the model resolution (Egbert et al., 2004). Here we show the effects of improving horizontal resolutions from 1/10° to 1/32°

on the global barotropic tide simulations of FIO-COM. We also present the simulated global tide in EXP2, which is a global baroclinic tide model.

For the 1/10° barotropic model, the tidal amplitude is smaller than TPXO9 in some regions such as west of Panama and the northeast Pacific, while the tidal amplitude is stronger than TPXO9 in the Labrador, Okhotsk and Andaman seas (Figures 8b, c). These biases are obviously reduced in the 1/32° barotropic model (Figure 8d), which indicates that the tidal

dissipation in the regions of complex terrain is better resolved as the horizontal resolution is increased. The RMSE distributions show that the tidal accuracy of the 1/32° barotropic tide model is improved almost everywhere over that of the 1/10° barotropic model (Figure 8e). The statistics show that for the $M_2$ constituent, the global averaged RMSE of tidal



elevation is decreased from 9.65 cm to 8.06 cm, reduced by 16.4%. Here, the tidal elevation RMSE is calculated for regions with water depth exceeding 1000 m and located between 65ºS and 65ºN following previous publications (Arbic et al., 2004;

Egbert et al., 2004).

The simulated global tide of EXP2, which is a global baroclinic tide model, is shown in Figure 8f. The overall pattern agrees well with the TPXO9. The topography drag scheme is turned off in EXP2, as a result, the simulated tidal amplitude is significantly larger than that of the barotropic experiments (Figures 8b, d) with topography drag optimally tuned, especially in the Atlantic Ocean and the east Pacific. The global averaged tidal elevation RMSE is 16.1 cm almost double of the

optimally tuned experiments. While the tidal amplitude in the western Pacific region agree well with that of TPXO9, the amplitude bias and RMSE is much smaller than that of Atlantic Ocean. This result may indicate that a considerable portion of tidal energy conversion in the western Pacific region may be explicitly resolved in the EXP2. Since the goals of this study are focused on the investigation of tide-circulation coupled processes, we believe that the global tide accuracy is sufficient to support this purpose.

In order to investigate whether the tide of 1/32° model could interact more strongly with the circulation than the 1/10° model, the incoherent tide is calculated and compared between the 1/10° and 1/32° models (EXP2) in Figure 9. Firstly, the coherent tidal constants of 8 main tidal constituents ($M_2$, $S_2$, $N_2$, $K_2$, $K_1$, $O_1$, $P_1$, and $Q_1$ ) are obtained by applying harmonic analysis to the hourly SSH model outputs. Secondly, the incoherent signals are obtained by extracting the predicted coherent signals of the 8 main tidal constituents. Finally, a Butterworth 10th order band pass filter with semi-diurnal (1.73 − 2.13 cpd)

bands is adopted to calculate the incoherent tide time series. The incoherent tide amplitude is defined as the standard deviation of the incoherent tide time series. The incoherent tide amplitude of the 1/32° model is obviously stronger than that of 1/10° model. The increased incoherent tide amplitude in the 1/32° model should be attributed to the increased eddy activities as indicated in the EKE comparisons (Figure 1).

**3.2 Effects of surface wave-tide-circulation coupling**

Prior to examining the effects of surface wave-induced mixing in the newly established FIO-COM32 model, the accuracy of the ocean wave model is evaluated. Figure 10 shows the comparisons of Jason3 and model simulated significant wave height (SWH). From the comparison, the model results are interpolated onto the satellite ground tracks. Both the distributions and seasonal cycle can be well reproduced by the global 1/4° resolution online coupled FIO-COM. The RMSE of the simulated SWH against Jason3 along track geophysical data records is calculated. The Jason3 data are produced and

distributed by Archiving, Validation, and Interpolation of Satellite Oceanographic data (AVISO, http://www.aviso.altimetry.fr/en/home.html). We use the low-pass filtered Jason3 data for the comparison, the global mean RMSE of the simulated SWH is 0.57 m.

The ocean upper mixed layer is a crucial layer locating between ocean interior and atmosphere. The MLD determines the heat and momentum content of the upper ocean boundary layer, and is a key factor that reflects the ability of an ocean

model. Figure 11 shows the derived summer MLD based on Argo observations (a), EXP1 (b) and EXP2 (c) simulations. The





summer is the mean of July, August and September (JAS) for the Northern Hemisphere, and the mean of January, February and March (JFM) for the Southern Hemisphere, respectively. The MLD is defined as the depth at which potential density increases by 0.125 kg/m³ from the corresponding surface values. The model results are interpolated onto the Argo profiles in both space and time.

Both EXP1 and EXP2 are able to reproduce the general patterns of summer MLD distributions. However, the MLD of EXP1 without Bv is shallower than that of Argo observations in most of the ex-tropical regions (Figure 11d). These shallow biases are significantly alleviated in EXP2 (Figure 11e), which is due to the inclusion of Bv as previous works showed. The comparison of the zonal mean MLD shows great improvement of EXP2 over EXP1. EXP2 fits quite well with the observations except in the Antarctic Circumpolar Current regions. The mean error of MLD in mid-latitude (10º-40º N/S)

decreases from -4.8 m (EXP1) to -0.6 m (EXP2). Although the resolution is increased to 1/32°, the Bv is still an important contributor that improves the summer MLD simulations. The impact of Bv in 1/10° model is quite similar to that of 1/32° (not shown), which means increasing horizontal resolution alone does not solve the long-standing MLD shallow biases in summer. As Bv deepens the simulated summer MLD, we further investigate whether it has a detectable effect on the length scale of the surface mixed layer instability (MLI) ($L_{sml}$). We follow Dong et al. (2020b) to calculate the $L_{sml}$, which is

defined as:

$$L_{sml} = 6 \frac{N_{ML} H_{ML}}{f}$$

where the $\mathbf{N_{ML}}$ is the depth averaged buoyancy frequency of the mixed layer, and $\mathbf{H_{ML}}$ is the MLD determined by the relative variance methods (Huang et al., 2018). As shown in Figure 12, compared with the Argo data, the $L_{sml}$ of EXP1 with the KPP vertical mixing scheme has an negative bias in most regions which is similar to that of Dong et al. (2020b), and the $L_{sml}$ of EXP2 with Bv is improved and more conformed with that of Argo data (Figure 12d).

As the model resolution is increased and global tide is explicitly included in EXP2, the model is able to simulate global internal tide (IT), which is activated when tidal currents flow over rough topography. Since a large portion of barotropic to baroclinic energy transfer can be explicitly resolved in EXP2, we disable the parameterized topography drag that has been used in the barotropic model. Figure 13 compares the global internal tide fields of satellite MOIST (Multivariate Inversion of Ocean Surface Topography) (Ubelmann et al., 2021) and model simulations. We apply the radial filter with the radius of 4º

as high-pass filter to remove the barotropic tide signals in the open ocean in the model SSH output. Then, harmonic analysis of 1-year high-passed SSH time series is performed to obtain the simulated $M_2$ internal tide amplitude. As shown in Figure 13, for both the $M_2$ and $K_1$ constituents, the MOIST and EXP2 agree well with each other in the positions of the IT "hot spot" generation sites, and the long-range propagation patterns. For example, the well-known generation site of $M_2$ IT near the Hawaii, and world's strongest $K_1$ IT near Luzon strait are well reproduced in the model. The model simulations tend to

be more energetic than the MOIST data, and whether this bias is due to the mesoscale contamination of the satellite data in the MOIST or insufficient IT related energy dissipation in present model remains to be explored in the future. The globally




averaged bias of amplitude of M$_2$ internal tide at the sea surface is 0.7 cm. Some published global baroclinic tide models such as HYCOM (Arbic et al., 2012; Shriver et al., 2012) show more conformed internal tide amplitude compared with the satellite observations than this paper. This may due to that the topographic drag used in their model
damps both the amplitude of barotropic and internal tide, while we do not use such kind of dissipation scheme in EXP2.

Previous studies (SK12, CX17) reported that the high-resolution model simulated SSH wave number spectral slope in the 70-250 km mesoscale domain is quite different from that of satellite observations. In the previous models, the slopes range between -5 and -4 in both low to high latitude regions. While the satellite observations show strong latitudinal variability, with much flatter slope of about -1 in tropical ocean to about -5 and -4 in high latitude and high EKE regions.
CX21 showed that internal tide induced SSH signals is a critical factor influencing the wave number spectral slopes. With tide included, the wave number spectral slopes in the 70-250 km mesoscale range of their 1/50° Atlantic regional ocean model show much better agreement with that of satellite observations than the model results without tide. Figure 13 shows that the internal tide induced sea surface undulations with spatial scale of tens to hundreds kilometres can reach several centimetres, so it is needed to explore the different responses of wave number spectral slopes in the EXP1 and EXP2.
The SSH wave number spectral slope of 70-250 km range is calculated as follows: Firstly, the model SSH data are interpolated onto Jason3 ground tracks. Secondly, the spectral slopes are calculated on a 1° x 1° grid for the global ocean. For each grid point, tracks falling into a 10° x 10° super-domain centered at the grid point are taken for spectral calculation. In addition, it also requires sub-tracks are longer than 500 km and have 90% or more good quality data. Prior to spectral analysis, the trend of each sub-track is removed, and a Hanning window is applied. The spectra is obtained by averaging all
the SSH spectra obtained via a fast Fourier transform (FFT). Finally, the spectral slope is calculated in the 70-250 km mesoscale range, which is in line with Xu and Fu (2012).

Figures 14a, b and c show SSH wave number spectral slope of 70-250 km range of Jason3 along track filtered data for EXP1 and EXP2. Figure 14a resembles to the results of Xu and Fu (2012), which shows strong latitudinal variability of wave number spectral slopes of the satellite observations. Figure 14b shows the wave number spectral slopes of EXP1 has much
weaker latitudinal variability, which resembles to that of SK12 and CX17. While the wave number spectral slopes of EXP2 show substantially improved agreement with that of satellite observations shown in Figure 14a. The flattened wave number spectral slopes of EXP2 in low EKE regions result from internal tide induced SSH undulations which can be clearly observed in SSH snapshots (Figures 14g and h, further in a Hovmöller diagram in Figure A-5). These internal tide signals have spatial scale of tens to hundreds of kilometres, and become nontrivial where the background eddy and circulation field
is relatively weak. Figures 14d, e and f show wave number spectra at three representative sites. Three locations are chosen for representing typical conditions: A, high EKE and active internal tides; B, typical equatorial regions; and C, high EKE but inactive internal tides. In the high latitude and high EKE regions, the spectra is less affected by the internal tide signals. However, in the low latitude and low EKE regions, the presence of internal tide substantially alters the spectra and yield much more realistic simulations. The textures of SSH of EXP1 and EXP 2 shown in Figures 14g and h manifest many





differences which mainly result from the internal tide signals, and the EXP2 would be much closer to reality. Therefore, for high resolution global ocean models, tide-circulation coupling is important for more detailed and realistic SSH simulations.

To further explore how the tide-circulation coupling and the increased resolution affect the simulations of different processes (including the rotational processes of sub-mesoscale turbulence and divergent processes of inertial gravity waves, IGWs), the wave number-frequency spectra of SSH variance are calculated from the hourly outputs of EXP1, EXP2, and

EXP2Low simulations. Focusing on the high frequency sub-mesoscale processes, the wave number-frequency spectra are calculated every 30 days and average the results to obtain a spectral estimate. Before calculating the Discrete Fourier Transform (DFT), we remove linear trends and multiply the data by a three-dimensional Hanning window. Meanwhile, the dispersion curves of IGWs are estimated from the World Ocean Atlas 2013 climatological temperature-salinity profiles (Qiu et al., 2018).

With respect to sub-mesoscale IGWs, it displaces by clear discrete beams above the tenth vertical normal mode curve in EXP2 (Figures 15a-c), especially in the subtropical gyre of the western Pacific, which is invisible in EXP1 (Figures 15d-f). By comparing the SSH wave number-frequency spectrum of EXP1 and EXP2, it shows that after including tidal forcing, SSH wave number-frequency spectra are more energetic in the near inertial, diurnal, and semi-diurnal tidal frequency bands as well as along the dispersion curves of discrete vertical modes of inertial-gravity waves. This can further explain that the

substantially enhanced unbalanced motions are responsible for the better match of EXP2 to Jason3 than EXP1 in Figure 14. As shown in Figures 15a-b, in the regions with active IGWs, the energy distribution extends continuously from sub-mesoscale regime to the IGWs regime which may indicates a strong nonlinear interaction between them. While in the low resolution experiments shown in Figures 15g-h, there is either a gap between regimes of sub-mesoscale and IGWs or insufficient sub-mesoscale activities, which may limit the nonlinear interaction between them. It is obvious that, with the

increasing resolution, SSH variance density is more energetic for the sub-mesoscale (<50 km and below the dispersion relation curve associated with the tenth vertical normal mode).

## 4 Summary and discussions

In this paper, for the first time, a global 1/32° surface wave-tide-circulation fully coupled model of FIO-COM32 is developed and validated. Increasing the model horizontal resolution from 1/10° to 1/32° leads to significant improvements in

the simulated surface EKE, main paths of the Kuroshio and Gulf Stream, and global tide. The ICRE is proposed as a quantitative criteria to evaluate the simulated main paths of Kuroshio and Gulf Stream. As the model horizontal resolution increases from 1/10° to 1/32°, the ICRE of the simulated Kuroshio (Gulf Stream) path is decreased from $3.01 \times 10^{11}$ $(4.85 \times 10^{11})$ m² to $1.73 \times 10^{11}$ $(3.27 \times 10^{11})$ m². The RMSE of the EKE between the model simulations and satellite observations decreased from 63.5 $\text{cm}^2\text{s}^{-2}$ for the 1/10° model to 31.7 $\text{cm}^2\text{s}^{-2}$ for the 1/32° model. The global barotropic $M_2$ RMSE

decreased from 9.65 cm for the 1/10° model to 8.06 cm for the 1/32° model. Although the resolution is increased to 1/32°, the non-breaking wave induced mixing (Bv) is still a key factor in improving the summer MLD simulation against the Argo





observations, with the mean error of the mid-latitude summer MLD reduced from -4.8 m in the numerical experiment without Bv to -0.6 m with Bv. Internal tides can be explicitly simulated in this new model, the global comparisons of along-track SSH wave number spectral slopes in 70-250 km domain show that the internal tide induced SSH undulations is critical

to substantially improved agreement between the global 1/32° model and satellite data.

In our following works, the effects of Sea Surface Currents (SSC) on surface wave simulation need to be taken into consideration for a wave-circulation fully coupled model, which has shown to have important impact on the surface wave simulation (Ardhuin et al., 2017). As has been stated in this paper, the surface wave-ocean circulation online coupled version of the FIO-COM32 is ready to be implemented when the demand of the computational resources are met in the future, and

the effects of ocean current on the surface wave will be explored in depth. Figure 16 shows an example of the effect of SSC on the simulated significant wave height. The hourly SSC output from the FIO-COM32 in the western Pacific is fed into the MASNUM wave model to test SSC effects on surface wave. It is clear that the SSC has notable influences on the simulated significant wave height especially near Kuroshio. In MASNUM wave model, the SSC influence the surface wave simulations through advection, refraction and the energy source function due to interaction between surface wave and SSC

shear (Yang et al., 2005).

Since the internal tide can be resolved in the FIO-COM32 model, the more robust representation of energy dissipation of internal tide propagation is still an open question that needs to be addressed in the future. A proper dissipation scheme of the internal tide and its adaption with the traditional viscosity schemes for OGCM is still a daunting challenge. We have conducted numerical experiments (not shown) with an inordinately large background vertical viscosity of 1.0 m$^2$/s, and in

these experiments the amplitude of the simulated internal tide is brought down to the comparable level of the MOIST observations. These preliminary explorations indicate that a proper scheme of vertical mixing considering the internal gravity wave dissipation in the tide-circulation coupled ocean model need to be developed. At the same time, more proper representation of the unresolved barotropic to baroclinic tidal conversion needs to be considered in the future. More realistic treatments of SAL will benefit the accuracy of global barotropic tide models (Arbic et al., 2004).

A better prediction of ocean environmental parameters is the inexhaustible momentum of the ocean model community. As the ocean, especially the upper ocean, plays a dominant role in the climate system, ocean model improvement can shed light on new generation climate model development. Surface waves, tides and ocean circulations are often separately simulated by different numerical models. In this paper, we clearly show surface wave-tide-circulation coupling can dramatically improve our simulations. So, it is time for us to regard the ocean as a fully coupled dynamic system through

turbulence, and develop surface wave-internal wave-tide-circulation fully coupled models as the direction of new generation ocean model development.



## Code and data availability

The exact version of the model, input data used to produce the results in this paper and data to produce the plots are archived on Zenodo (https://doi.org/10.5281/zenodo.6221095), the minimal scripts to compile and run the model can be accessed on
Github (https://github.com/mom-ocean/MOM5). The atmosphere forcing data is downloaded from https://www.nco.ncep.noaa.gov/pmb/products/gfs/.

## Author contributions

BX was responsible for model development and drafted the manuscript; FQ designed the road map of this research and supervised the paper writing. All other authors contributed through discussion, data analysis and the paper writing.

## Competing interests

The authors declare that they have no conflict of interest.

## Acknowledgements

This research is jointly supported by the National Natural Science Foundation of China under Grant 41821004 and the Marine S&T Fund of Shandong Province for Pilot National Laboratory for Marine Science and Technology (Qingdao) (No.
2018SDKJ0106-1). This work is a contribution to the UN Decade of Ocean Science for Sustainable Development (2021-2030) through both the Decade Collaborative Centre on Ocean-Climate nexus and Coordination amongst decade implementing partners in P. R. China (DCC-OCC) and the approved Programme of the Ocean to climate Seamless Forecasting system (OSF).

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







**Figure 1 Eddy kinetic energy (EKE) of CMEMS from all satellite merged grided data (a), FIO-COM10 (EXP1Low) (b), and FIO-COM32 (EXP1) (c) model results.**








**Figure 2 Time series of globally averaged EKE (a), EKE of the 1/32° model filtered onto a 1/4° grid by a low-pass spatial filter with radius of ~50 km (b), and the EKE residual as high-pass component (c).**





**Figure 3 Mean SSH (left column) and SSH STD (right column) based on altimeter observations of CMEMS (a, b), FIO-COM10 (c, d) and FIO-COM32 (e, f) simulations.**




**Figure 4 Mean SSH of the Kuroshio region. The route of Kuroshio is represented by the contour line at level of 0.16 m, CMEMS (a), 1/10° model (EXP2Low) (b) and its circulation route errors (d), 1/32° model (EXP2) (c) and its circulation route errors (e). Note that the regional averaged value is extracted from the SSH before the ICRE analysis is conducted to ensure the SSH of the model and observation are comparable.**






**Figure 5 Same as Figure 4 but for the Gulf Stream region. Note that the ICRE of Gulf stream re-circulation is masked out to focus on the main path of Gulf Stream.**




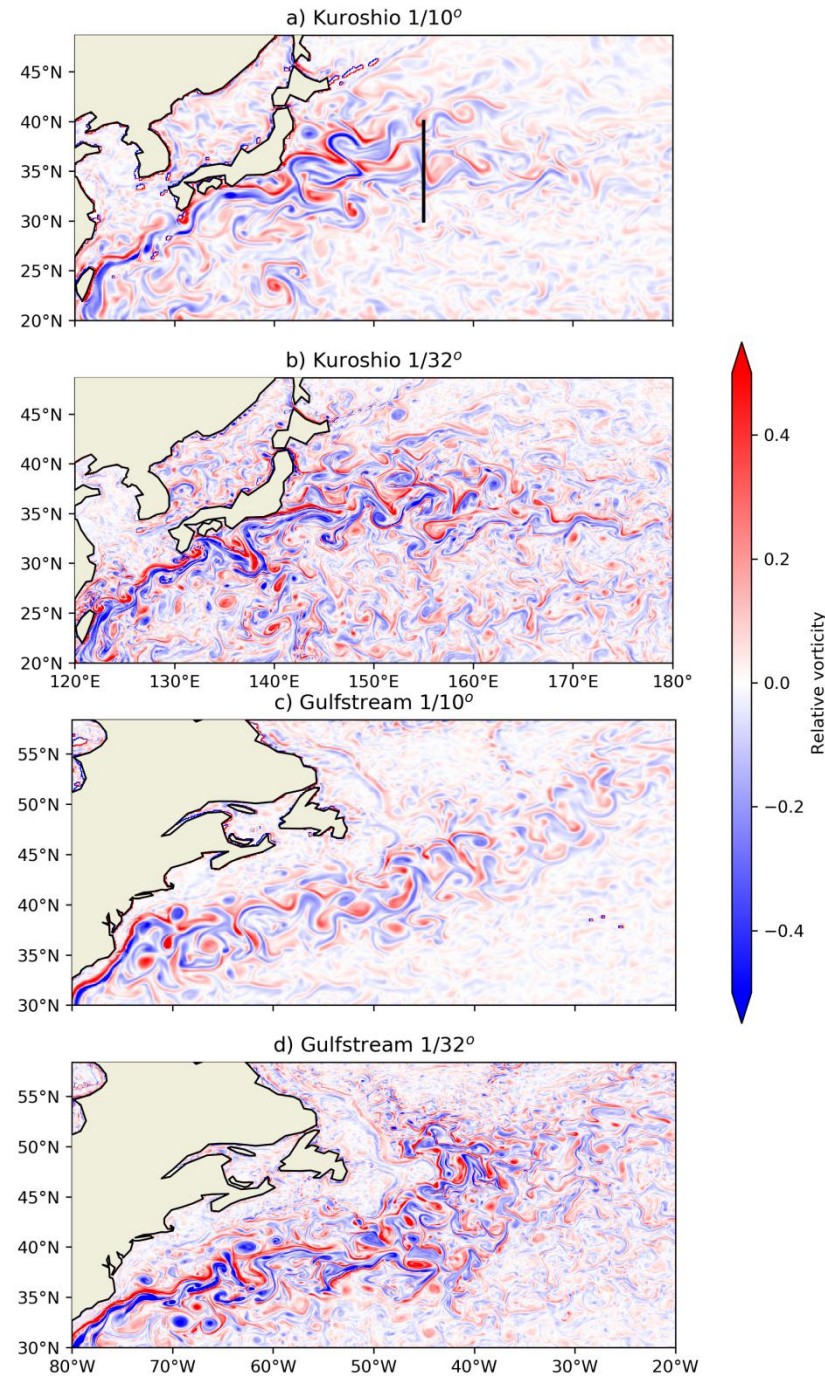

**Figure 6 Relative vorticity ($\dfrac{\zeta}{f}$) of sea surface current of 1/10° (a, c) and 1/32° (b, d) model results, snapshots of summer (1 September 2019) of the Kuroshio region (a, b), and that of the Gulf Stream region (c, d), respectively.**






**Figure 7 Same as Figure 6 but for winter (1 March 2019).**




**Figure 8 M₂ co-tidal charts of TPXO9 (a), barotropic tide model of 1/10° (b) and 1/32°(d) with topographic drag parameterisation best tuned, baroclinic tide model of 1/32° (EXP2) (f) without topographic drag parameterisation, and their RMSE errors (c, e and g).**

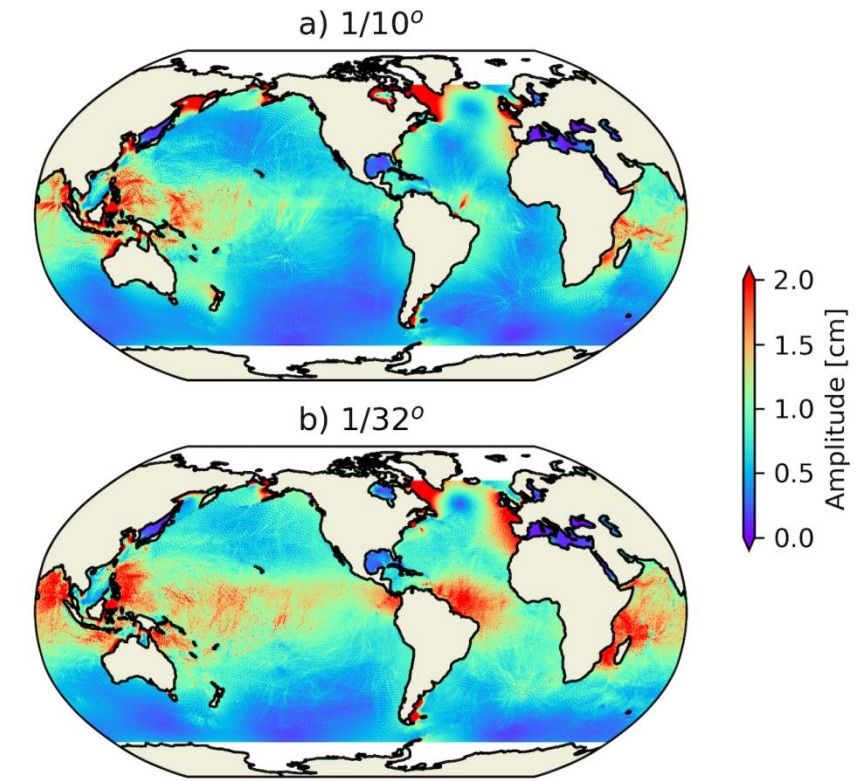


**Figure 9 Incoherent tide amplitude of semi-diurnal tidal band of 1/10° (a) and 1/32° (b) model.**

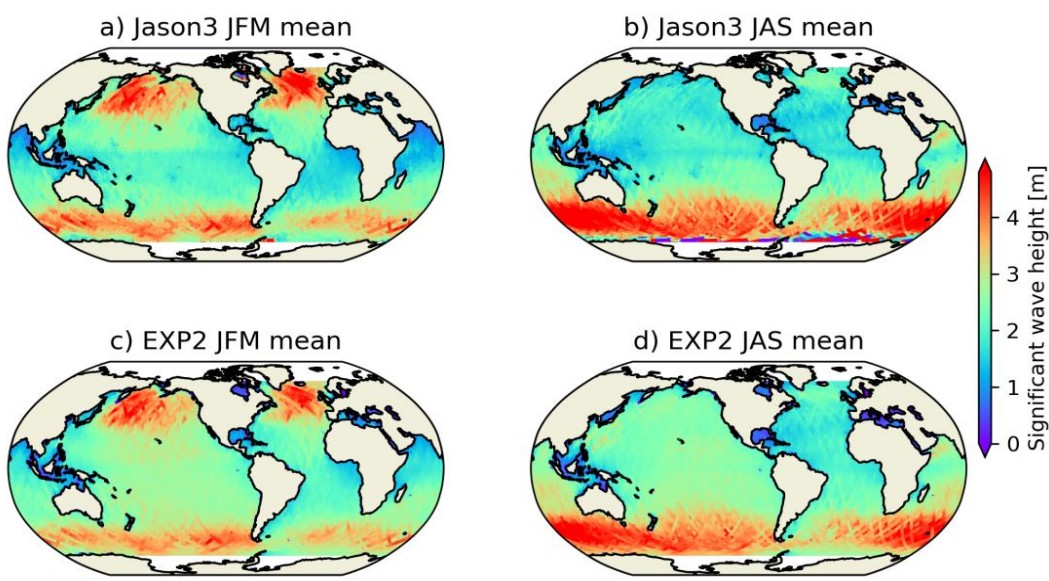

**Figure 10 Along track seasonal mean significant wave height of Jason3 (a, b) and EXP2 (c, d). Note that the surface wave model of**
665                                 **EXP2 is offline with horizontal resolution of 1/4°.**



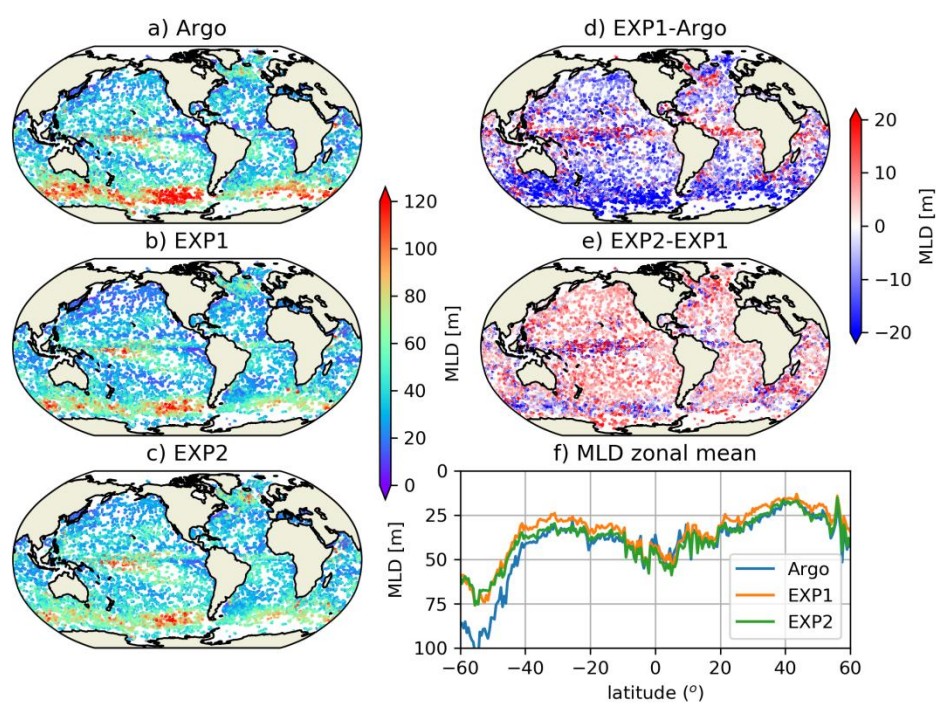

**Figure 11 Summer (JAS and JFM for Northern and Southern Hemispheres, respectively) MLD based on Argo observations (a), EXP1 (b) and EXP2 (c) simulations. The differences of MLD between EXP1 and Argo, EXP2 and EXP1, and the zonal mean MLD are shown in (d), (e) and (f), respectively.**


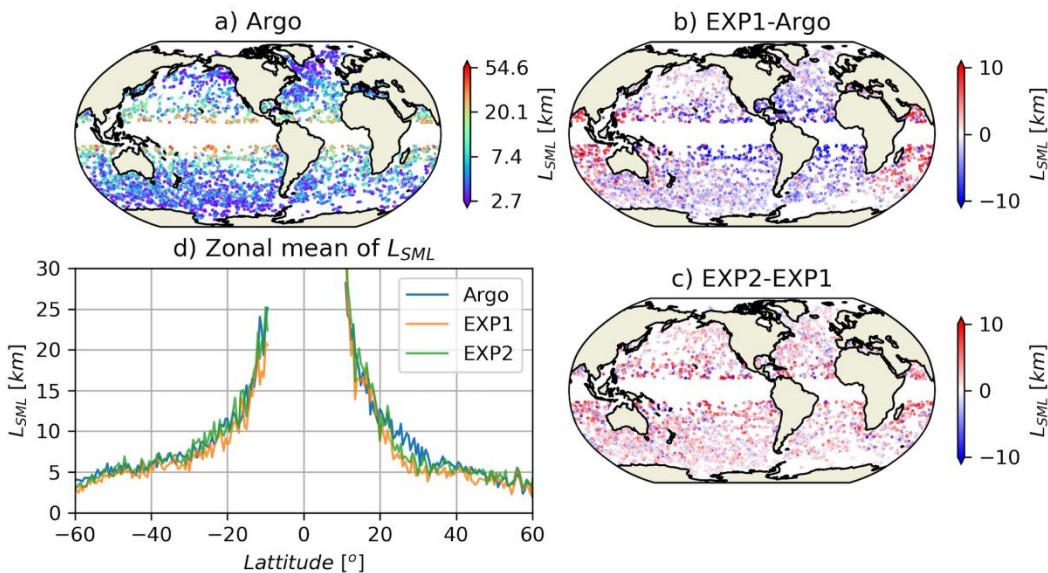





**Figure 12 Length scale of summer (JAS and JFM for Northern and Southern Hemispheres, respectively) surface mixed layer instability ($L_{sml}$) of Argo observations (a). The differences of $L_{sml}$ between EXP1 and Argo, EXP2 and EXP1, and the zonal mean MLD are shown in (b), (c) and (d), respectively. The mean error of $L_{sml}$ decrease from -0.96 km of EXP1 to 0.05 km of EXP2.**


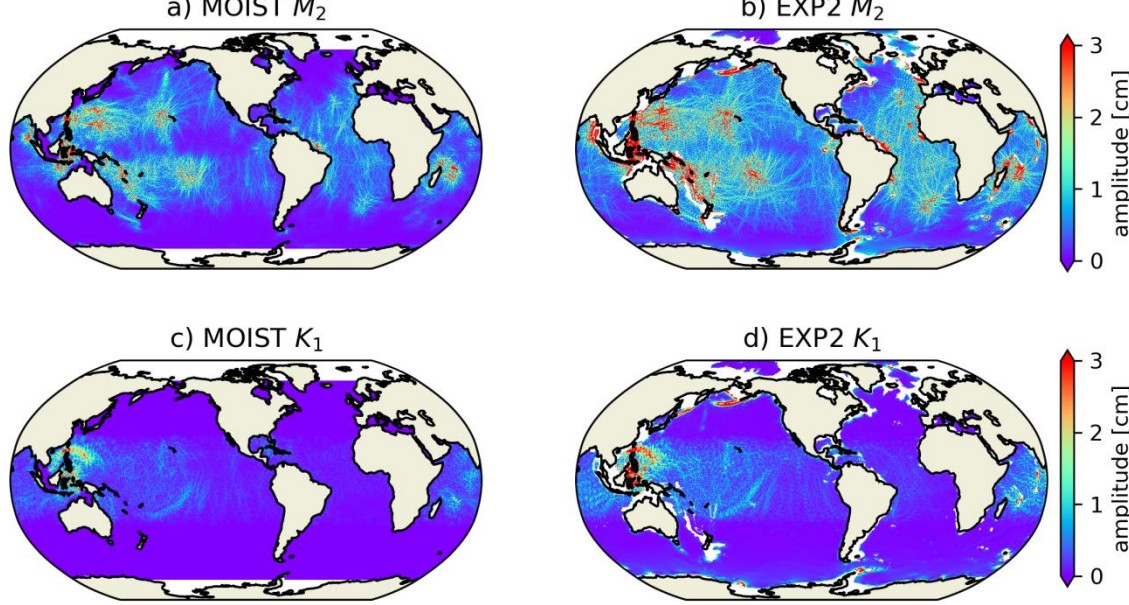

**Figure 13 $M_2$ internal tide amplitude at the surface from MOIST (a), EXP2 (b); (c) and (d) are the same as (a) and (b) but for $K_1$.**






**Figure 14 SSH wavenumber spectrum slope of 70-250 km of Jason 3 along track data (a), EXP1 (b) and EXP2 (c). SSH wave number spectrum at three sites in regions of Kuroshio extension (d), tropical Pacific Ocean (e) and Southern Ocean (f), the black lines denote the wavenumber of 70 and 250 km. Snapshots of SSH of EXP1 (g) and EXP2 (h) on 1 December 2018 in the rectangle region are shown in (a). Note that the sign of the spectrum slope is reversed to make them positive in a-c.**







**Figure 15 SSH variance spectrum amplitude[m²/(cpkm × cph)] in horizontal wave number-frequency space estimated from 10° × 10° boxes centered at (a-b) (138°E, 26°N) (c-d) (175°E, 0°) and (e-f) (175°E, 55°S). The upper and middle panels are based on hourly outputs of EXP2 and EXP1, respectively, and the lower are based on the 1/10° EXP2Low. Dashed white lines denote the inertial frequencies, and dashed black lines denote the tidal constituents of $S_2$, $M_2$, $K_1$, $O_1$. Solid black curves denote the dispersion relations for inertia-gravity waves of the first and tenth vertical modes. Note that the color scale is exponential.**



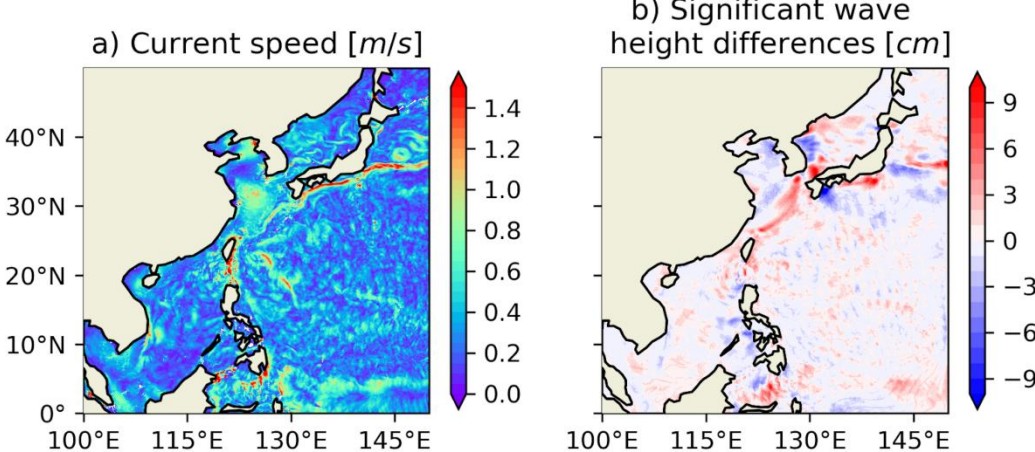

**Figure 16 Sea surface current (SSC) snapshot of 8 October 2018 (a), the differences of significant wave height between MASNUM wave models with and without SSC effects.**


**Appendix**

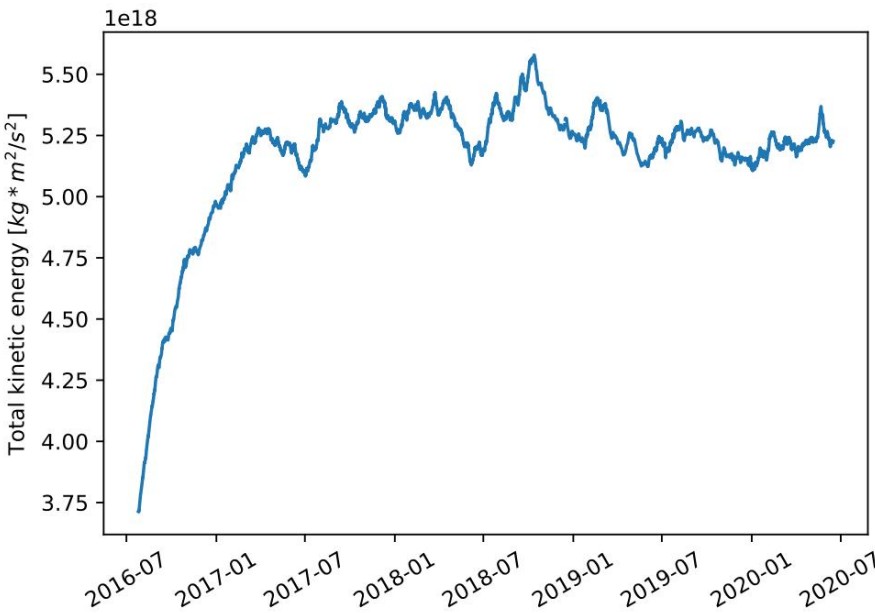


**Figure A-1 Time series of simulated total kinetic energy of EXP1.**





Figure A-1 shows the total kinetic energy of EXP1 since the beginning of the numerical experiment. The 1/32° model restarts from a 1/10° model states and enters a new steady state after a swift adjusting period of about 1 year. After the short spin-up period, the total kinetic energy shows a quite steady fluctuation.


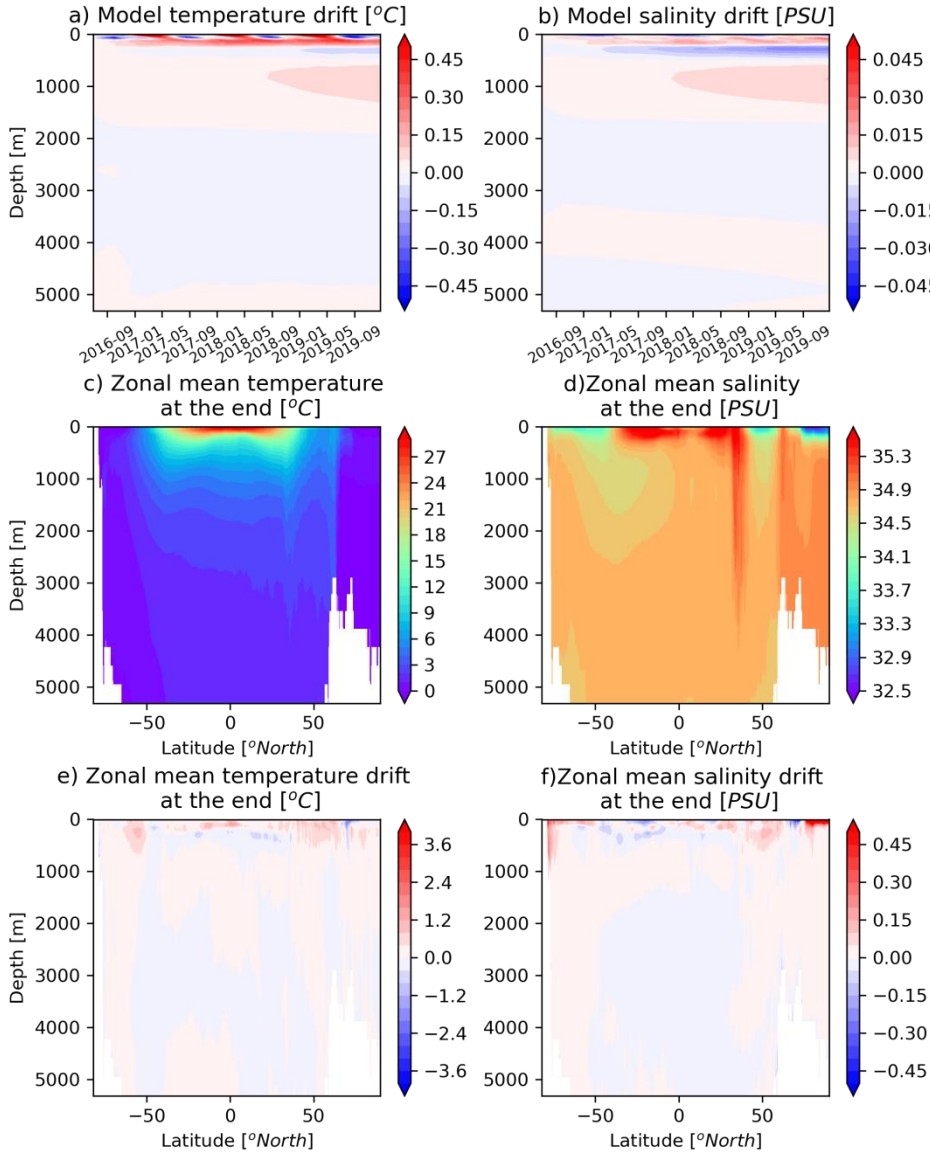

**Figure A-2 model drift of temperature (a) and salinity (b), zonal averaged temperature (c), salinity (d) and the corresponding drift (e, f) at the end of EXP1. As the last day of model integration is 31 Dec. 2019, so the drifts in (e, f) are calculated using simulation difference between 31 Dec. 2016 and 31 Dec. 2019.**



Since the time span of the numerical experiments is not long (3.5 years), the model drift is generally quite small. The model drift of temperature shows a strong seasonal cycle in the upper ocean and a warming trend in the sub-surface layer, the maximum drift of temperature at the end is about 0.2℃ (Figure A-2a). The upper 150 m becomes slightly salty and the sub-surface layer shows freshening drift, the salinity drift value is generally small, less than 0.02 PSU (Figure A-2b). The zonal mean temperature and salinity and their corresponding drifts are displayed in Figure A-2c-f, the zonal mean drifts are

moderate, and reasonable distributions of  temperature and salinity at the end are maintained.

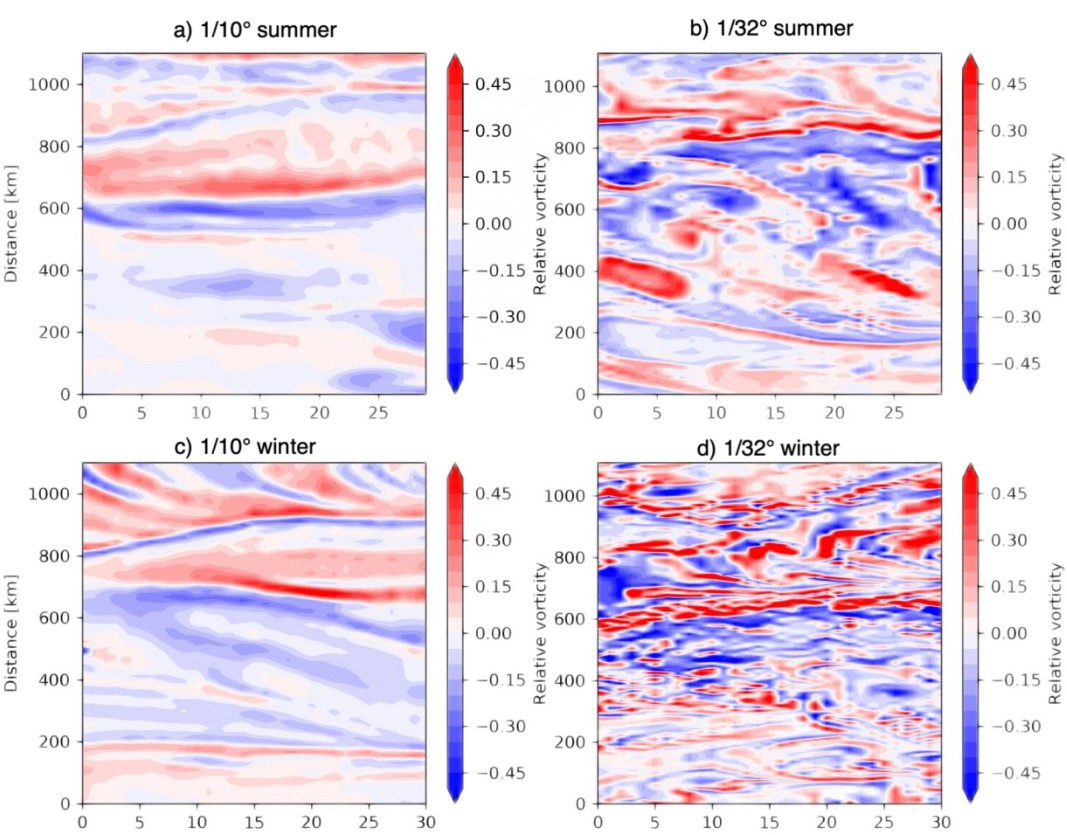

**Figure A-3 Hovmöller diagram of relative vorticity along a section in Kuroshio extension region, marked as black line shown in Figure 6.**





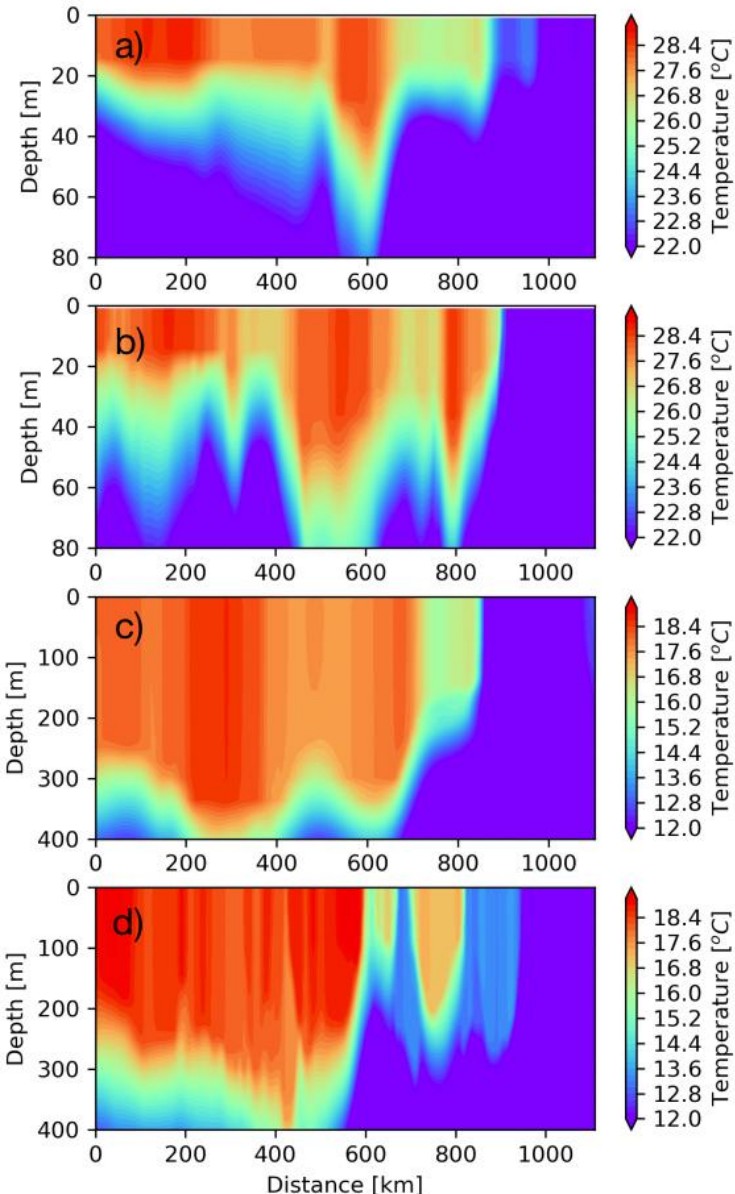

**Figure A-4 Temperature along the section in Kuroshio extension region of 1/10° (a, c) and 1/32° (b, d) model during summer (a, b) and winter (c, d) respectively.**

Figures A-3 and A-4 show the evolution of relative vorticiy and temperature along a typical section in Kuroshio extension region. The typical horizontal scale of resolved fronts in 1/10 ° model is about 60 km, and the horizontal scale in 1/32 ° model is about 3 times smaller ~20 km. The vertical scales of the resolved fronts of both 1/10 ° and 1/32 ° model are quite similar, the summer (winter) has a vertical scale of about 30 m (300 m).



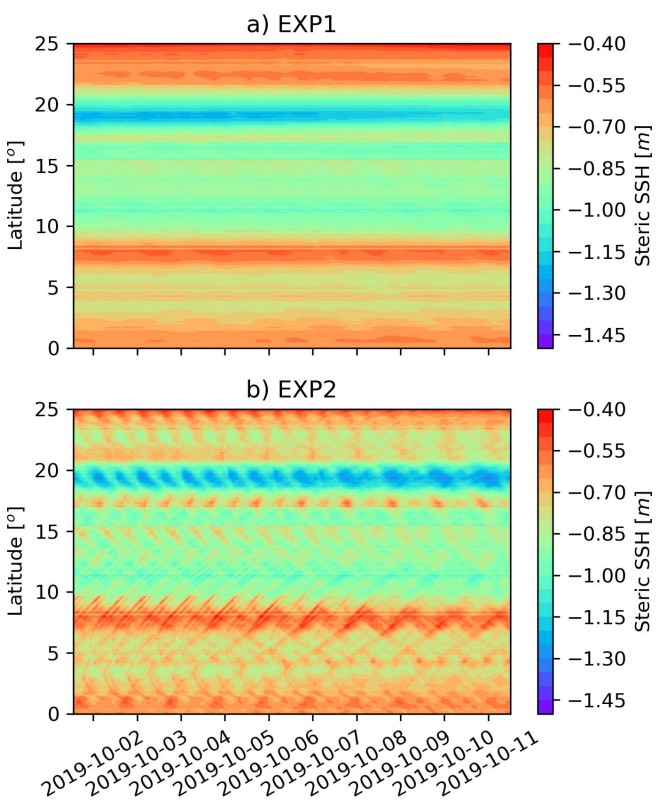

**Figure A-5 Hovmöller diagram of steric SSH along the section of 135° E.**

Figure A-5 shows the evolution of steric SSH along a section of 135 ° E. EXP2 with tide shows significant signals of the
propagation of internal tide. This figure together with Figures 14g-h explain how the internal tide signature affect the
simulated SSH.