# Peer review of "Development and validation of a global 1/32° surface wave-tidecirculation coupled ocean model: FIO-COM32"

_Geoscientific Model Development, 2022_

## Author Response (AR2)

We have carefully considered review comments and revised the manuscript accordingly. Point-to-point responses to the review comments from the referees are as follow.

Referee comments are in blue colored fonts, and our replies in black.

**Response to Referee Comments 1**

This manuscript describes the implementation and initial results of simulations using a very high-resolution (1/32°) global ocean model, including waves and tides. This is an exceptional effort and adds to a small handful of similar very high-resolution simulations of the ocean which have been undertaken to date. As such, it is suitable for publication in GMD and deserves to be eventually published.

However, the manuscript is essentially the same as their previous version, gmd-2022-52, which I have reviewed twice and was ultimately rejected. The main problem with their previous version, and the current submission, is their persistent belief that the Bv parameterisation represents mixing by non-breaking waves, which it does not. Although they have removed three of the previous references to Bv as representing mixing by non-breaking waves, there are still several places in which this belief is retained (which I will detail below). Unfortunately, therefore, the paper must be rejected again.

Firstly, though, I include again my response to the authors' previous comments about Bv: "I thank the authors for their response on the Bv parameterisation. However, no good reason has still been provided to support their assumption that w' and l' are in phase for the waves (to leading order). Even though, yes, the fluid motion is not fully irrotational, to first order, for a monochromatic wave train, w' and l' are in quadrature, so that $\langle w'l' \rangle = 0$. However, in Qiao et al. (2010, Ocean Dynamics 60: 1339-1355), a single monochromatic wave is considered, and the key underlying assumption for Bv is made between equations 34 and 35 that w' and l' are in phase, so that $\langle w'l' \rangle$ is NON-ZERO. There is no justification given for this, either in the paper, or the authors' response to my original point on this. Regarding the wave tank observations which are purported to support Bv, I would need to look at these very closely as a separate exercise, but would make the observation that mixing effects will result from the sides of the tank which may be difficult to allow for. And their third point that Bv has already been used in a range of leading models and can dramatically improve their mixed layers is irrelevant to the point in question: of course, the addition of a (possibly large) near-surface mixing term will result in the reduction of over-heating in the ocean surface through additional downward mixing." The overall effect of Bv is to add an arbitrary, unphysical mixing term (which could be quite large) to the upper ocean.

The places which need to be corrected in the present manuscript, concerning Bv, are now:

l. 16. "The non-breaking surface wave-induced mixing (Bv) is proven to still be" should read "A previously described upper ocean mixing scheme (Bv) is proven to still be"

l. 252-270. Why is the discussion of the Stokes shear force introduced here, what

impact does it have on anything being discussed? i.e. what does the epsilon parameter compare the Stokes shear force to? Is this intended to justify the inclusion of Bv in the model (ie by saying this will be important when the Stokes shear force is important)? Note that Bv does NOT represent the effect of mixing by non-breaking waves. Therefore, I cannot see the point of this discussion about the Stokes shear force, and the discussion in lines 252-270 should be deleted.

l. 305. "Prior to examining the effects of surface wave-induced mixing in the …" we are NOT examining the effects of surface-wave induced mixing here, only of the Bv parameterisation which does NOT represent breaking by non-breaking waves. This sentence must be changed to "Prior to examining the effects of the Bv mixing scheme in the …"

l. 411. Bv does NOT represent mixing by non-breaking waves, so change "the non-breaking wave induced mixing (Bv)" to "the mixing induced by Bv"

**Author response:**

After careful study of the comments, we are responding as follows:

First, we would  like to thank the reviewer for your overall impression of our manuscript, I quote: "This manuscript describes the implementation and initial results of simulations using a very high-resolution (1/32°) global ocean model, including waves and tides. This is an exceptional effort and adds to a small handful of similar very high-resolution simulations of the ocean which have been undertaken to date. As such, it is suitable for publication in GMD and deserves to be eventually published."

However, you also had persistent and specific misgivings expressed in multiple comments. Let us summarize the main concerns as follows:

1. Your main concern is our view on the Bv parameterization representing mixing by non-breaking waves, which you think it does not.
2. You also thinks that "the Stokes shear force introduced here, what impact does it have on anything being discussed? …. Note that Bv does NOT represent the effect of mixing by non-breaking waves. Therefore, I cannot see the point of this discussion about the Stokes shear force."
3. "We are NOT examining the effects of surface-wave induced mixing here, only of the Bv parameterisation which does NOT represent breaking by non-breaking waves."

We can conclude that the reviewer does not object to the wave induced or enhanced mixing, but you are specifically objecting to the point that "non-breaking wave" could induce or enhance mixing.

It is obvious that ocean mixing involves turbulence. One of the main sources of oceanic turbulence is from breaking waves as reported by the classic studies by Thorpe (2005). However, they both pointed out that the turbulence so generated is

confined to the upper most layer of the ocean with the thickness of the order of the wave amplitude. The main contribution of Bv is to propose a mechanism on how the turbulence, generated either by breaking waves or surface drift shear instability, propagates down, in a stably stratified fluid, to a depth that would influence the large-scale general circulation.

Mixing is an energy problem. Therefore, the role of the surface waves, being the most energetic motion on the ocean upper layer, should be seriously considered. This problem has been a hot subject for investigation as reviewed by Qiao et al (2016). This problem was the initiated by Phillips (1961). After many studies, the best effort was the detailed analysis by Teixeira and Belcher (2002, referred as TB thereafter). Phillips first proposed the wave orbital velocity could induce turbulence straining. TB carefully formulate the interaction mechanism using "**a single monochromatic wave**" as a model. The main conclusion of TB can be summarized (with some direct quotations) as follows:

"*A rapid-distortion model is developed to investigate the interaction of weak*

*turbulence with a monochromatic irrotational surface water wave. The model is applicable when the orbital velocity of the wave is larger than the turbulence intensity, and when the slope of the wave is sufficiently high that the straining of the turbulence by the wave dominates over the straining of the turbulence by itself. The turbulence suffers two distortions. Firstly, vorticity in the turbulence is modulated by the wave orbital motions, which leads to the streamwise Reynolds stress attaining maxima at the wave crests and minima at the wave troughs; the Reynolds stress normal to the free surface develops minima at the wave crests and maxima at the troughs (see Figure 6). Secondly, over several wave cycles **the Stokes drift** associated with the wave tilts vertical vorticity into the horizontal direction, subsequently stretching it into elongated streamwise vortices, which come to dominate the flow (see Figure 8). These results are shown to be strikingly different from turbulence distorted by a mean shear flow, when 'streaky structures' of high and low streamwise velocity fluctuations develop.*" The predicted vorticity distribution has been confirmed by open ocean turbulence measurement by Qiao et al (2016).

"*The kinetical energy generation from the turbulence and wave interaction is estimated as*

$$\frac{\partial E_K}{\partial t} \approx -2\overline{u_1 u_3} a_w^2 k_w^2 \sigma_w \approx -\overline{u_1 u_3}\frac{du_s}{dx_3}(x_3 = 0). \qquad (3.16)$$

*This estimate of TKE production has a similar form to the term involving the **Stokes drift** in the TKE equation (5.1) of McWilliams et al. (1997). It is as if there were a Stokes drift 'shear' that generates TKE.*

[Figure]

*"Figure 6. Schematic diagram showing the vorticity stretching and compression induced by the orbital motion at the crest and at the trough of a surface wave, in a frame of reference travelling with the wave.*

*"The distortion of the turbulence by **the Stokes drift** becomes clear after a considerable number of wave cycles.*

[Figure]

*"Figure 8. Schematic diagram showing the tilting and stretching of the vertical vorticity carried out by **the Stokes drift** of a surface wave over a number of wave cycles, in a fixed frame of reference.*

*"The physical mechanism for the intensification of the streamwise vortices in the present model is the same as mechanism CL2 of Craik & Leibovich (1976) for the generation of Langmuir circulations. It involves the tilting of vertical vorticity by*

*__the Stokes drift__ of the wave and its amplification as streamwise vorticity (figure 8). The difference is that the Craik–Leibovich formulation departs from an infinitesimal vertical vorticity perturbation arising from transverse variations of the wind-induced shear current, whereas in the present model, there is initially a finite and isotropic distribution of vorticity, associated with the turbulence. In both cases, __the Stokes drift__ selectively amplifies the vertical vorticity component as streamwise vorticity."*

Clearly, the Stokes drift is the key element for wave-turbulence interactions. In fact, Stokes drift exists in non-breaking, inviscid, irrotational monochromatic wave train. Its existence, however, is in Lagrangian sense, in a particle following frame to reveal mass-transport, first derived by Stokes in 1847, and hence the eponymous drift. From the dynamic point of view, the existence and the influence of Stokes drift is very clear: Even irrotational and non-breaking wave causes energy, $E$, propagation. In wave motion,

$$E=Mc,$$

Where $E$ is the energy, $M$ is the momentum and $c$ the phase velocity. Momentum is mass multiplies by mass-transport velocity. As $c$ is large, the momentum associated with wave motion is small, of a second order magnitude, but not zero, even the energy is large. In the TB study, the wave is a monochromatic, irrotational, and non-breaking surface water wave. For the key role is **the selective amplification by the Stokes drift** (a.k.a. Stokes shear) on the long-term straining of the turbulence vertical vorticity to the horizontal direction, thus induce and enhance vertical mixing. There should be no doubt the wave involved is
> Non-breaking,
> Even monochromatic, and
> Stokes drift plays the key role.

Our contribution in the parameterization of Bv is to express its effect in term of wave parameters. In parameterization of Bv, we selected a velocity scale and a length scale. The product of these quantities gives the "Eddy Viscosity" equivalence. In ocean dynamic, many phenomena involved are clearly unknown. Parameterization is the only way to move forward. For example, the eddy viscosity in the general circulation used is just to make the computation stable, with little physical justification. I wish the reviewer would take this point into consideration too.

Next, let us discuss the Bv parameterization scheme, which is also of a major concern of the reviewer. On this point, you stated: "I thank the authors for their response on the Bv parameterisation. However, no good reason has still been provided to support their assumption that w' and l' are in phase for the waves (to leading order). Even

though, yes, the fluid motion is not fully irrotational, to first order, for a monochromatic wave train, w' and l' are in quadrature, so that <w'l'> = 0. However, in Qiao et al. (2010, Ocean Dynamics 60: 1339-1355), a single monochromatic wave is considered, and the key underlying assumption for Bv is made between equations 34 and 35 that w' and l' are in phase, so that <w'l'> is NON-ZERO. There is no justification given for this, either in the paper, or the authors' response to my original point on this. Regarding the wave tank observations which are purported to support Bv, I would need to look at these very closely as a separate exercise, but would make the observation that mixing effects will result from the sides of the tank which may be difficult to allow for. And their third point that Bv has already been used in a range of leading models and can dramatically improve their mixed layers is irrelevant to the point in question: of course, the addition of a (possibly large) near-surface mixing term will result in the reduction of over-heating in the ocean surface through additional downward mixing." The overall effect of Bv is to add an arbitrary, unphysical mixing term (which could be quite large) to the upper ocean.

The discrepancy on the parameterization scheme should be easily resolved. The key is whether "for a monochromatic wave train, w' and l' are in quadrature, so that <w'l'> = 0," or "w' and l' are in phase, so that <w'l'> is NON-ZERO." In the original Qiao et al (2010) paper, there is no <w'l'> term. Instead, the term should be $<w'_{3w}l_{3w}>$. The difference looks subtle, but it is critical. In Qiao et al (2010), it was carefully stated, and we quote that

> "We use an analogy to **the Prandtl mixing length theory to parameterize** the momentum mixing induced by wave motion. ... For ocean surface wave processes, **we assume that the mixing length $l_{iw}$ is proportional to the range of the particle displacement in the i-th direction**. We need to note that the concern of the expression **$u'_{iw}$ here is not the mathematical derivation, but a concept and assumption of equivalent scales**. $u'_{iw}$ should be understood as **the increment of the wave motion velocity at the spatial interval of $l_{iw}$** in the i-th direction."

Therefore, $u'_{iw}$ is precisely the Stokes drift velocity, and $l_{iw}$ is the particle excursion range. They are both scales for the wave motion, NOT MATHEMATICAL DERIVATION; obviously, there is no phase relationship at all. "The increment of the wave motion velocity at the spatial interval of $l_{iw}$" is clearly in the Lagrangian sense, or the Stokes drift. As stated above, Stokes drift exists for non-breaking, irrotational, and even Monochromatic water waves.

NB: Please also note the difference between $\tau_{wwij}$ and $\tau_{wcij}$: These represent two types of Reynolds stresses: $\tau_{wwij}$ are wave related stresses, the pure wave quantities and could be derived mathematically. For irrotational wave field, their values are zero when the velocities are in quadrature. $\tau_{wcij}$ are wave current interactions that include

the wave induced Stokes drift and ambient current field. We will concentrate in the wave related Stokes drift here, which exists only in Lagrangian frame. As Qiao et al (2010) was formulated in Eulerian frame, Stokes drift could not be derived and could only be parameterized. However, this much is clearly stated: " $u'_{iw}$ should be understood as the increment of the wave motion velocity at the spatial interval of $l_{iw}$ in the i-th direction;" therefore, the velocity is exactly the Stokes drift.

Now, let us turn to some of the minor points:

1. This is NOT a fully-coupled model as the surface waves are computed offline and do not interact with the ocean circulation fields as they evolve, so change "surface wave-tide-circulation fully coupled model" to "surface wave-tide-circulation coupled model".

We set up a fully coupled model. Due to the limitation of computer resources, we run the wave model offline. Following your review comments, we remove the word "fully".

2. Regarding the wave tank observations which are purported to support Bv, I would need to look at these very closely as a separate exercise, but would make the observation that mixing effects will result from the sides of the tank which may be difficult to allow for.

The effect of side of the tank was a critical issue when that paper was first submitted for publication. The order of magnitude argument could help here. It suffices to say that the molecular viscosity is many orders smaller than the Bv. Additionally, other experiments on the dissipation rate measurement in the open ocean (i.e. Sutherland, 2013) also support the validity of Bv. There are no sides in the ocean, of course.

We thank the reviewer in pointing out many detailed syntax and expression anomalies. We have followed all these suggestions in most places.

Finally, we hope these explanations on the non-breaking, irrotational and even monochromatic wave could contribute to enhancing mixing would resolve the reviewer's concerns.

Reference:

Huang, C. J., Qiao, F., Dai, D., Ma, H., & Guo, J., 2012: Field measurement of upper ocean turbulence dissipation associated with wave-turbulence interaction in the South China Sea, J. Geophys. Res., 117, C00J09, doi:10.1029/2011JC007806

Phillips, O.M., 1961: A note on the turbulence generated by gravity waves. J Geophys Res 66:2889–2893. doi:10.1029/JZ066i009p02889

Qiao F, Yuan Y, Deng J, Dai D, Song Z., 2016 Wave–turbulence interaction-induced vertical mixing and its effects in ocean and climate models. Phil. Trans. R. Soc. A 374:20150201. http://dx.doi.org/10.1098/rsta.2015.0201

Sutherland G., Ward, B. & Christensen, K. H., 2013: Wave-turbulence scaling in the ocean mixed layer. Ocean Sci., 9, 597–608, doi:10.5194/os-9-597-2013

Teixeira, M. A. C. And Belcher, S. E., 2002: On the distortion of turbulence by a progressive surface wave. J. Fluid Mech. 458, 229-267

Thorpe, S. A., 2005: The Turbulent Ocean. Cambridge University Press, New York

Other comments which should be addressed are as follows:

l. 22. Need to explain what are the "unbalanced motions" referred to here? Are they the motions induced by internal tides for instance?

**Author response:** As introduced by Rocha et al. (2016) and Chereskin et al. (2019), the "unbalanced motions" referred to the geostrophically unbalanced motions, i.e. ageostrophic motions, e.g. internal tides and inertia-gravity waves. This has been added in the revision in line 22.

Reference:

Chereskin, T. K., Rocha, C. B.,Gille, S. T., Menemenlis, D., and Passaro, M.: Characterizing thetransition from balanced to unbalanced motions in the southern California Current, Journal of Geophysical Research: Oceans, 124, 2088–2109. doi:https://doi.org/10.1029/2018JC014583, 2019

Rocha, C. B., Chereskin, T. K., Gille, S. T., and Menemenlis, D.: Mesoscale to submesoscale wavenumber spectra in Drake Passage, J. Phys. Oceanogr., 46(2), 601-620, doi:10.1175/JPO-D-15-0087.1, 2016.

l. 121-125: In the high resolution case, the wave and ocean circulation model are coupled offline, so the wave field cannot interact with the ocean circulation fields as they change (ie because the wave fields are previously saved as fixed data files). Need to explain this fully here.

**Author response:** The explanation has been added in the revision. As shown in Fig. 16b, the model is surface wave and ocean circulation models fully coupled, and the ocean current can modulate the surface wave height. For computer efficiency, we turn off the coupling pro tempore.

l. 138-139. The Bv field applied to the high-resolution (1/32°) model is calculated from an online-coupled lower resolution model (1/4°). The lower resolution model will have different circulation fields (i.e. the currents will be slower and broader, and probably in different places), so the Bv field applied to the high-resolution model will not be appropriate. What difference will this make to the high-resolution results?

**Author response:** We totally agree with you that the same resolution of surface wave and ocean circulation models is the best choice. However, in current stage, we have difficulty to run a 1/32° wave model due to insufficient computational resources. Our group tested daily and monthly averaged Bv in a coarse resolution ocean circulation model, the difference is not big (Zhao et al., 2012), and both can much improve the upper ocean simulation. So, we can deduce that the coarse resolution Bv of 1/4° can work for FIO-COM32, although not perfect.

Reference:

Zhao, C., Qiao, F., Xia, C., and Wang, G.: Sensitive study of the long and short surface wave-induced vertical mixing in a wave-circulation coupled model, Acta Oceanologica Sinica, 31(4), 1-10, doi:10.1007/s13131-012-0215-y, 2012.

l. 160 The wave-tide-circulation model is NOT fully coupled since the waves are run offline – change this to "In EXP2, wave-tide-circulation coupling is enabled".

**Author response:** Done.

p. 8 and fig.s 4 and 5. What longitudes are the ICRE diagnostics defined over (presumably those in the figures)?

**Author response:** The longitudes of the ICRE diagnostics are defined as those in the Figures 4 and 5, i.e., 115° E-160° E and 90° W-45° W respectively. The explanation is added in line 250 of the revision.

p. 8 and fig. 5. How is the ICRE defined for the Gulf Stream in the 1/10° model, since the contour used for its definition does not exist eastwards of about 63°W?

**Author response:** The ICRE defined for the Gulf Stream in the 1/10° model is identical to that of 1/32° model, hence the total area misfit eastwards of about 63°W of the 1/10° model is quite large due to that the 1/10° model is not able to reproduce the deep penetration of the Gulf Stream into the Atlantic ocean. As a result, in Fig. 5 the ICRE of the 1/10° model is much larger than that of 1/32° model.

l. 293-294: how do you justify the claim that "the global tide accuracy is sufficient to support" …. " the investigation of tide-circulation coupled processes" given that the errors in fig 8g are in excess of 25cm over large regions of the ocean?

**Author response:** As shown in Fig. 8f, the overall pattern of the $M_2$ global tide agrees well with the TPXO9, most of the amphidromic points in the open basins are reproduced reasonably. On the other hand, the barotropic tide is filtered out in the wavenumber spectrum analysis to avoid the direct impacts of the barotropic tide. We have refined the sentences (lines 300-315 of the revision) accordingly.

The deteriorating of the accuracy of the global baroclinic tide model is an open question for ocean model community. Since a large portion of tidal energy conversion is resolved by the model, the tidal dissipation parameterization for the 1/32° baroclinic run must be able to distinguish the unresolved and resolved tidal energy dissipation/conversion. To our knowledge, this kind of tidal dissipation scheme still does not exist in the existing publications and is a daunting challenge for the ocean model scientific community. Hence we choose to disable the topographic drag scheme used in the barotropic runs not to tune it, which is aligned with LLC4320 of MITgcm (Arbic et al., 2018).

We have also conducted additional numerical experiments to test the validity of our model physics, and to further illustrate the advantages of our modelling strategy. Following Arbic et al. (2010), we implement topographic drag scheme (Jayne and St. Laurent, 2001) in the global baroclinic run to improve the accuracy of the simulated global tide, a 25-hour boxcar time filter is adopted in the topographic drag scheme and the pressure force to separate the tidal and non-tidal motions. In our implementation the topographic drag scheme is applied in the barotropic momentum equation. Due to limited time, the best tuned experiments are conducted in the 1/10° model, while a short-term experiment is conducted in the 1/32° model. Three numerical experiments based on the 1/10° model are conducted: 1, barotropic run with topographic drag tuned (BT_WithDrag), 2, baroclinic run without topographic drag (BC_NoDrag), and 3, baroclinic run with topographic drag best tuned (BC_WithDrag). All of the three experiments run for 6 months, and the outputs of last 3 months are used to conduct harmonic analysis. Figure R1 shows the co-tidal charts and RMSE of three numerical experiments, the simulated M2 tide of BC_WithDrag is significantly improved compared to that of BC_NoDrag. The RMSE of BC_WithDrag is even smaller than that of BT_WithDrag. The global averaged RMSE of tidal elevation is decreased from 17.25 cm of BC_NoDrag to 8.8 cm of BC_WithDrag, which has similar accuracy with that of 8.26 cm of Arbic et al. (2010). The above results provide us a solution to improve tide simulation.

Although the simulated tide of numerical experiment without topographic drag is less accurate, in current stage it still has the following two substantial advantages over the experiment with topographic drag:

1, As mentioned by Arbic et al. (2010), the 25-hour boxcar time filter adopted in the numerical experiment with topographic drag is "lagged in time (i.e. from the previous 25 h)" and is not able to separate the tidal and non-tidal motions exactly. While in the numerical experiment without topographic drag, the 25-hour boxcar time filter is no

longer needed, which exempt the degradation of the ocean circulation simulation caused by the time filter. The satisfying simulation of main paths of Kuroshio and Gulf Stream in the EXP2 (Figure 4c and 5c of the revision) might be an evidence supporting this advantage.

2, The 25-hour boxcar time filter excludes most of the convergent/divergent barotropic flows caused by internal tides, which degrades the dynamic sea level to "surface tide plus circulation simulations" (Figure R2). This is essentially decoupling between surface tide and internal tide. While in the experiment without topographic drag, the tide-circulation coupled processes (such as internal tides and related inertial gravity waves) are faithfully represented in the dynamic sea level, which is the foundation of the wavenumber spectrum analysis in this paper (Figures 14 of the revision).

We hope the above added numerical experiments and explanations could alleviate reviewer's concerns about the tidal accuracy.

[Figure]

Figure R1 $M_2$ co-tidal charts of TPXO9 (a), barotropic (b) and baroclinic (f) tide model of 1/10°  with topographic drag scheme best tuned, baroclinic tide model of 1/10° without topographic drag (d), and their RMSE (c, g and e).

[Figure]

Figure R2 SSH snapshot of the 1/32° model (EXP2) with 25-hour boxcar time filtered topographic drag (a), and without topographic drag (b).

Reference:

Arbic, B. K., Wallcraft, A. J., and Metzger, E. J.: Concurrent simulation of the eddying general circulation and tides in a global ocean model, Ocean Model., 32(3), 175-187, 2010.

Arbic, B., Alford, M., Ansong, J., Buijsman, M., Ciotti, R., Farrar, J., Hallberg, R., Henze, C., Hill, C., Luecke, C., Menemenlis, D., Metzger, E., Müeller, M., Nelson, A., Nelson, B., Ngodock, H., Ponte, R., Richman, J., Savage, A., and Zhao, Z.: A primer on global internal tide and internal gravity wave continuum modeling in HYCOM and MITgcm, https://doi.org/10.17125/gov2018.ch13, 2018.

Jayne, S. R., and St Laurent, L. C.: Parameterizing tidal dissipation over rough topography, Geophys. Res. Lett., 28(5), 811-814, 2001.

Fig. 12 (d) shows the $L_{SML}$ not the MLD as specified in the caption.

**Author response:** Thanks for pointing out this, the caption is corrected in the revision.

Fig.s 13 and 14 and discussion of the inclusion of internal tides. This was nice to see and the most useful part of the paper. The inclusion of the internal tides appears to add SSH variability between 70-250 km and increase the amount of energy in the spectra (fig. 14) in the more quiescent tropical regions, so that the spectral slope is reduced and more in agreement with the observations. However, fig. 13 clearly shows that the internal tidal field at the surface is too strong, probably because of the lack of bottom dissipation. Can the authors comment on how to reconcile these two aspects, ie if the internal tidal field was realistic, what would the effect on the spectral slopes be (e.g. in fig. 14(e)).

**Author response:** Although the slopes of EXP2 (Fig. 14c) are more in agreement with the observations than that of EXP1 (Fig. 14b), there is still some discrepancies, especially in the low latitude regions of Atlantic. In this region, the slope of EXP2 is

apparently more flat than the observations, which may indicate the internal tide here is too strong. More realistic internal tidal field may further improve the agreement of the model and observations. This also remind us that a proper dissipation scheme of the internal tide and its adaption with the traditional viscosity schemes for OGCM need to be developed in the future.

l. 390. Replace "it displaces by clear discrete beams" with "these are shown by clear discrete beams"

**Author response:** Thanks, it is replaced in the revision.

l.395: what are the unbalanced motions – presumably the internal tides, IGWs etc?

**Author response:** The "unbalanced motions" referred to the ageostrophic motions, here mainly internal tides and inertia-gravity waves. The explanation has been added in the revision.

Fig. 15. Please say which solid black line is the tenth normal mode, and which is the first?

**Author response:** The caption has been modified. "Solid black curves denote the dispersion relations for inertia-gravity waves of the first (upper) and tenth (lower) vertical modes. "

Fig. 15 caption is wrong: e.g the box centred at 138°E, 26°N is shown in panels (a), (b) and (c) and not in panels (a) and (b) as in the caption, with similar comments for the other rows of panels.

**Author response:** Thanks for the comments, the caption has been corrected in the revision.

l. 402. This is NOT a fully-coupled model as the surface waves are computed offline and do not interact with the ocean circulation fields as they evolve, so change "surface wave-tide-circulation fully coupled model" to "surface wave-tide-circulation coupled model".

**Author response:** This has been changed in the revision.

l. 438-439: "we clearly show surface wave-tide-circulation coupling can dramatically improve our simulations" is not true since the surface waves are not fully coupled. So delete the word "clearly".

**Author response:** Done.

Minor corrections to the English (up to line 147) are as follows (there are many more such corrections which could be made, so a thorough read-through by a native English speaker would be of benefit here):

l. 33 Further improved resolution has a significant impact

l. 48 The most uncertain term

l. 51 proposed an upper ocean mixing scheme of Bv

l. 61 in many coarse resolution

l. 64 coarse and high resolution

l. 97 configurations and design

l. 147 baroclinic experiments so that

**Author response:** Thanks for these corrections, they have been corrected in the revision. We have turned to a native English speaker to polish the draft.

Referee comments are in blue colored fonts, and our replies in black.

**Response to Referee Comments 2**

The paper introduces 1/32° resolution simulations with waves and tides. I think it is a nice first effort in terms of model development. However, I don't find the results from these runs warrant a solid scientific publication. Therefore, I recommend rejection to the paper.

The conclusion on the effect of resolutions is hardly innovative: higher resolution resolves smaller scale processes, and that affects eddy-rich systems like Kuroshio and Gulf Stream. In contrast, ACC seems to get worse in 1/32°, which seems to me is a more interesting aspect, but the authors curiously left it out.

**Author response:** Thank you so much for pointing out that this is "a nice first effort in terms of model development" regarding the 1/32° resolution simulations with waves and tides in this paper. For responding your comments: "The conclusion on the effect of resolutions is hardly innovative: higher resolution resolves smaller scale processes, and that affects eddy-rich systems like Kuroshio and Gulf Stream", we clarify/repeat the novelty of this submission as follows:

1. It is the first try in the world for model development to couple the surface wave, tide and circulation into a high resolution model, which is emphasized by reviewer, and the model performs quite well. This innovation on ocean model development should deserve the publication on GMD. We also would like to quote the comments of another referee "adds to a small handful of similar very high-resolution simulations of the ocean which have been undertaken to date."

2. For the first time, we quantitatively demonstrated that: (1) By including tide, the generated internal tide modulates the Sea Surface Height on global scale which is the key factor contributing to the substantially improved agreement of model against satellite observations in terms of wave number spectral slopes of mesoscale range in the low latitude and low EKE regions. To our knowledge, this is the first report for the global ocean model in improving the simulation of the wave number spectral slopes of mesoscale range. (2) The non-breaking surface wave-induced mixing (Bv) is proved still to be an important contributor that improves the agreement of the simulated summer mixed layer depth against the Argo observations even with high horizontal resolution of 1/32°, with the simulated MLD mean error reduced from -4.8 m without Bv to -0.6 m in experiment with Bv. We provide a solution for accurate MLD which is crucially important for the climate system, marine ecosystem, and Tropical Cyclone evolution.

3. As an example, we quantitatively assess the improvements of simulated EKE of the developed global ocean models with horizontal resolutions of 1/10° and 1/32°. Then

we quantitatively assess the impact of horizontal resolution on the simulated paths of Kuroshio and Gulf Stream, by proposing the Integrated Circulation Route Error (ICRE) as a quantitative criteria. As far as we know, it is also the first time that this ICRE criteria for ocean model evaluation is proposed and applied.

The ACC is not specifically studied in this paper, we will look into it in the future.

Based on all the above distinct innovations, and with serious consideration, we select this highly reputed journal of GMD.

The tide part is also confusing. First, the global averaged RMS errors in the barotropic runs are suspiciously high (even higher than some of the past works with lower resolutions), which begs further investigation of the validity of the model physics and makes it hard to argue the conclusions based on these simulations. Furthermore, the 1/32° baroclinic run yields even worse tides than not only its barotropic counterpart but also the 1/10° resolution runs. In my opinion, this result alone discredits the model and the conclusions from the simulations.

**Author response:** We disagree with the review comments of "the global averaged RMS errors in the barotropic runs are suspiciously high". As we can see in Figures 8d and e, the simulated co-tidal charts of barotropic runs agree pretty good with that of TPXO9, and the global averaged RMS errors in the barotropic runs is 8.06 cm. It is a very reasonable statistic and laid a solid foundation for the development of high resolution surface-wave-tide-circulation coupled model. Going through the previous publications with similar model physics (Shriver et al., 2012), we noticed that their simulation errors for $M_2$ is 7.48 cm, while our simulation error is 8.06 cm. For global barotropic tide models, the RMSE become insensitive to the model resolution beyond ~1/10° (Egbert et al., 2004). Although we still have room to improve the accuracy (such as, changing model discretization from B-grid to C-grid, more realistic treatment of Self Atracktion and Loading effect etc), to call it "suspiciously high ... begs further investigation of the validity of the model physics and makes it hard to argue the conclusions based on these simulations" is unfair and obviously an exaggeration.In this paper, we reported that the internal tide can modulate sea surface height, and serves as the key factor contributing to the substantially improved agreement of model against satellite observations in terms of wave number spectral slopes of mesoscale ranges. This is the first report for the global ocean model. If we are wrong, we ask Reviewer 2 to educate us. The key that we can observe the important tide-circulation coupled effects in our numerical experiments is that we have chosen the proper model settings, including normal model background viscosity, and closed topographic drag scheme used in the barotropic runs not to tune it, which is aligned with LLC4320 of MITgcm (Arbic et al., 2018). These modelling strategies pursue honest representation of the tide-circulation coupled processes, not just the accuracy of the tide. As shown in Fig. 8f, the pattern of the 1/32° baroclinic run

agrees well with that of TPXO9. Since a large portion of tidal energy conversion is resolved by the model, the tidal dissipation parameterization for the 1/32° baroclinic run must be able to distinguish the unresolved and resolved tidal energy dissipation/conversion. To our knowledge, this kind of tidal dissipation scheme still does not exist in the present publications and is a daunting challenge for the ocean model scientific community. If the reviewer knows some publication, please let us know. Hence we choose to disable the topographic drag scheme used in the barotropic runs not to tune it, which is aligned with LLC4320 of MITgcm (Arbic et al., 2018). Although this leads to larger errors in simulated global tide, we believe that in the current stage, the representation of tide-circulation coupled processes will benefit from this strategy just as shown in Figure 14c.

We have also conducted additional numerical experiments to test the validity of our model physics, and to further illustrate the advantages of our modelling strategy. Following Arbic et al. (2010), we implement topographic drag scheme (Jayne and St. Laurent, 2001) in the global baroclinic run to improve the accuracy of the simulated global tide, a 25-hour boxcar time filter is adopted in the topographic drag scheme and the pressure force to separate the tidal and non-tidal motions. In our implementation the topographic drag scheme is applied in the barotropic momentum equation. Due to limited time, the best tuned experiments are conducted in the 1/10° model, while a short-term experiment is conducted in the 1/32° model. Three numerical experiments based on the 1/10° model are conducted: 1, barotropic run with topographic drag tuned (BT_WithDrag), 2, baroclinic run without topographic drag (BC_NoDrag), and 3, baroclinic run with topographic drag best tuned (BC_WithDrag). All of the three experiments run for 6 months, and the outputs of last 3 months are used to conduct harmonic analysis. Figure R1 shows the co-tidal charts and RMSE of three numerical experiments, the simulated M2 tide of BC_WithDrag is significantly improved compared to that of BC_NoDrag. The RMSE of BC_WithDrag is even smaller than that of BT_WithDrag. The global averaged RMSE of tidal elevation is decreased from 17.25 cm of BC_NoDrag to 8.8 cm of BC_WithDrag, which has similar accuracy with that of 8.26 cm of Arbic et al. (2010). The above results provide us a solution to improve tide simulation.

Although the simulated tide of numerical experiment without topographic drag is less accurate, in current stage it still has the following two substantial advantages over the experiment with topographic drag:

1, As mentioned by Arbic et al. (2010), the 25-hour boxcar time filter adopted in the numerical experiment with topographic drag is "lagged in time (i.e. from the previous 25 h)" and is not able to separate the tidal and non-tidal motions exactly. While in the numerical experiment without topographic drag, the 25-hour boxcar time filter is no longer needed, which exempt the degradation of the ocean circulation simulation caused by the time filter. The satisfying simulation of main paths of Kuroshio and Gulf Stream in the EXP2 (Figure 4c and 5c of the revision) might be an evidence supporting this advantage.

2, The 25-hour boxcar time filter excludes most of the convergent/divergent barotropic flows caused by internal tides, which degrades the dynamic sea level to "surface tide plus circulation simulations" (Figure R2). This is essentially decoupling between surface tide and internal tide. While in the experiment without topographic drag, the tide-circulation coupled processes (such as internal tides and related inertial gravity waves) are faithfully represented in the dynamic sea level, which is the foundation of the wavenumber spectrum analysis in this paper (Figures 14 of the revision).

We hope the above added numerical experiments and explanations could alleviate reviewer's concerns about the tidal accuracy.

[Figure]

Figure R1 $M_2$ co-tidal charts of TPXO9 (a), barotropic (b) and baroclinic (f) tide model of 1/10° with topographic drag scheme best tuned, baroclinic tide model of 1/10° (d) without topographic drag, and their RMSE (c, g and e).

[Figure]

Figure R2 SSH snapshot of the 1/32° model (EXP2) with 25-hour boxcar time filtered topographic drag (a), and without topographic drag (b).

Reference:

Arbic, B. K., Wallcraft, A. J., and Metzger, E. J.: Concurrent simulation of the eddying general circulation and tides in a global ocean model, Ocean Model., 32(3), 175-187, 2010.

Arbic, B., Alford, M., Ansong, J., Buijsman, M., Ciotti, R., Farrar, J., Hallberg, R., Henze, C., Hill, C., Luecke, C., Menemenlis, D., Metzger, E., Müeller, M., Nelson, A., Nelson, B., Ngodock, H., Ponte, R., Richman, J., Savage, A., and Zhao, Z.: A Primer on Global Internal Tide and Internal Gravity Wave Continuum Modeling in HYCOM and MITgcm, https://doi.org/10.17125/gov2018.ch13, 2018.

Egbert, G. D., Ray, R. D., and Bills, B. G.: Numerical modeling of the global semidiurnal tide in the present day and in the last glacial maximum, J. Geophys. Res., 109(C3), doi:10.1029/2003JC001973, 2004.

Jayne, S. R., and St Laurent, L. C.: Parameterizing tidal dissipation over rough topography, Geophys. Res. Lett., 28(5), 811-814, 2001.

Shriver, J., Arbic, B. K., Richman, J., Ray, R., Metzger, E., Wallcraft, A., and Timko, P.: An evaluation of the barotropic and internal tides in a high-resolution global ocean circulation model, J. Geophys. Res., 117(C10), doi:10.1029/2012JC008170, 2012.

In addition, I find the draft poorly written. There are numerous grammar mistakes, and many sentences are either confusing or awkward.

**Author response:** We have invited a native English speaker to polish the draft. The certificate is attached here. The corresponding author is the co-editor-in-chief of Ocean Modelling, and have actively taken part in international cooperation including as the Decade Advisory Board member of United Nations Decade of Ocean Science for Sustainable Development. Our English is far from perfect or beautiful, but should

be okay for scientific understanding. If the reviewer could provide detailed comments on which sentence should be polished, that will be highly appreciated.

[Figure]

[Figure]

**Editing Certificate**

This document certifies that the manuscript

**The development and validation of a global 1/32° surface wave-tide-circulation coupled ocean model: FIO-COM32**

prepared by the authors

**Bin Xiao, Fangli Qiao, Qi Shu, Xunqiang Yin, Guansuo Wang, Shihong Wang**

was edited for proper English language, grammar, punctuation, spelling, and overall style by one or more of the highly qualified native English speaking editors at AJE.

This certificate was issued on **June 16, 2022** and may be verified on the AJE website using the verification code **4DOD-AB96-4A89-D93D-A356** .

Neither the research content nor the authors' intentions were altered in any way during the editing process. Documents receiving this certification should be English-ready for publication; however, the author has the ability to accept or reject our suggestions and changes. To verify the final AJE edited version, please visit our verification page at aje.com/certificate. If you have any questions or concerns about this edited document, please contact AJE at support@aje.com.

AJE provides a range of editing, translation, and manuscript services for researchers and publishers around the world. For more information about our company, services, and partner discounts, please visit aje.com.